# Reconfigurable two-dimensional optoelectronic devices enabled by local ferroelectric polarization

Liang Lv[1], Fuwei Zhuge [1], Fengjun Xie[1], Xujing Xiong[1], Qingfu Zhang[1], Nan Zhang[1], Yu Huang[1] & Tianyou Zhai [1]

Ferroelectric engineered pn doping in two-dimensional (2D) semiconductors hold essential promise in realizing customized functional devices in a reconfigurable manner. Here, we report the successful pn doping in molybdenum disulfide ($MoS_2$) optoelectronic device by local patterned ferroelectric polarization, and its configuration into lateral diode and npn bipolar phototransistors for photodetection from such a versatile playground. The lateral pn diode formed in this way manifests efficient self-powered detection by separating ~12% photo-generated electrons and holes. When polarized as bipolar phototransistor, the device is customized with a gain ~1000 by its transistor action, reaching the responsivity ~12 A W$^{-1}$ and detectivity over $10^{13}$ Jones while keeping a fast response speed within 20 μs. A promising pathway toward high performance optoelectronics is thus opened up based on local ferroelectric polarization coupled 2D semiconductors.

[1] State Key Laboratory of Materials Processing and Die & Mould Technology, School of Materials Science and Engineering, Huazhong University of Science and Technology, 430074 Wuhan, People's Republic of China. Correspondence and requests for materials should be addressed to F.Z. (email: zhugefw@hust.edu.cn) or to T.Z. (email: zhaity@hust.edu.cn)

Two-dimensional (2D) semiconductors have drawn extensive interests in functional electronic devices for their exotic optoelectronic properties by the quantum confinement in atomic thickness and feasibly changed characteristics under external modulation[1–5]. Using field effects coupled by high-$\kappa$ dielectrics[6,7], ionic liquids and gels[8,9], and ferroelectric (FE) polarization[10–12], their electric properties have been widely tuned from semiconductor to highly conductive metals and oppositely to the insulators. Such features have enriched the emerging of various gate-modulated devices[6,13], including transistors[14,15], logic inverters[7], memories[11,16], light-emitting diodes (LEDs)[17], and photodetectors[3,18,19]. An ultimate pursuit to this end would be however a reconfigurable functional device that can be customized on demand, so that a universal device architecture can be deployed in various applications. FE material-coupled 2D semiconductors hold special promise in reforming these devices toward a reconfiguration, given their large, non-volatile yet rewritable feature of remnant polarization[20–23]. There is an increasing interest in integrating FE materials, such as BiFeO$_3$[10], P(VDF-TrFE)[12], PbZrTiO$_3$[16,24], and LiNbO$_3$[25], to 2D functional devices. However, further efforts are, to a marked degree, hampered by the ability of FE polarization in controlling pn doping of 2D materials[23], which constitutes a fundamental building block for functional electronic devices[26,27].

Given the selective pn doping of 2D materials, a variety of fundamental devices that enable the digital technology can be developed, for example, pn diodes and amplifying bipolar transistors[28,29]. In optoelectronics, these devices based on pn junctions could manifest self-powered or fast and high gain photodetection beyond the usual photoconductors. However, early attempts to construct the kind of devices relied prominently on locally buried gates[28], lateral and vertical heterojunctions[30–33], the behavior of which are complicated to manipulate without fine-tuned pn doping. For example, the amplification gain in the resulting bipolar transistor is severely limited by the low carrier injection efficiency at the emitter–base junction, for example, based on MoS$_2$/WSe$_2$[29] or MoS$_2$/BP heterojunctions[34]. Moreover, these efforts fail to satisfy reconfigurable customization on demand with the predefined gate electrodes, epitaxial sequence in lateral junction, or stacking order in van der Waals heterostructures. With the large remnant polarization, FE coupling to 2D semiconductors is favored to realize reconfigurable optoelectronics based on rewritable pn doping in the 2D components, which however has not been demonstrated yet[10,12,16]. For example, organic FE materials, such as P(VDF-TrFE) copolymers, have in the past manifested indispensable merits in their facile spin-coating fabrication. Sophisticate top-gate integration methods have been adopted for their integration with various 2D semiconductors, including MoS$_2$[11,12,35,36], MoSe$_2$[37], MoTe$_2$, and WSe$_2$[38]. However, these efforts generally fail to achieve pn modulation even with the large remnant polarization of P(VDF-TrFE) ($\sim$8 $\mu$C cm$^{-2}$) over conventional gate oxides, which draw increasing demand to improve the FE integration in 2D optoelectronic devices for reconfigurable pn doping and functionalization.

Here, we report on reconfigurable optoelectronic photodetectors with FE polarization-defined pn doping in MoS$_2$ using the device configuration of bottom electrical contacts. With the rewritable FE polarization using a scanning atomic force microscope (AFM) tip, the MoS$_2$ photoconductor is facile customized into pn diodes and bipolar phototransistors with optimal photodetection performance. This allows us to demonstrate a lateral pn diode with an ideal factor of 1.7 for self-powered photodetection that separate $\sim$12% photogenerated electrons and holes, and a npn bipolar phototransistor with a gain $\sim$1000 with fine-adjusted pn polarization. Such reconfigurable device

characteristics may promote the evolvement of smart image sensors that respond to external light levels for balanced photoresponse gain and energy efficiency. Our study on reconfigurable optoelectronic devices using FE polarization thus unravels a pathway towards customizing novel functional optoelectronics based on 2D semiconductors.

## Results

**Reconfigurable pn doping in MoS$_2$ by FE polarization**. Figure 1a illustrates the configuration of FE-coupled MoS$_2$ device, in which an AFM tip is employed to switch the local polarization. To fabricate the device, multilayer MoS$_2$ from mechanical exfoliation is used as the semiconductor channel, while the FE copolymer P(VD-TrFE) is spin coated on the top with a thickness of $\sim$200 nm. The copolymer film is later crystallized into its orthorhombic $\beta$ phase by annealing at 135 °C for 15 min[39]. The coercive field of as-prepared P(VDF-TrFE) film, under which the FE polarization switches, is measured in agreement with literature to be $\sim$5 × 10$^7$ V m$^{-1}$ using capacitance–voltage measurement (Supplementary Fig. 1)[40]. The FE nature of P(VDF-TrFE) in intimate contact with MoS$_2$ is then exploited to engineer the carrier doping in MoS$_2$ through their reversible polarization by an external poling field. The representative coupling between the FE polarization in P(VDF-TrFE) and the adjacent MoS$_2$ is illustrated in Fig. 1b. To obtain p-type doping in MoS$_2$, upward polarization (P↑) in P(VDF-TrFE) is required, while reversely the opposite downward polarization (P↓) is desired to enhance n-type doping. Specifically, in order to achieve widely tuned doping in MoS$_2$, we adopt bottom electrode contacts by placing thin MoS$_2$ flakes on top of predefined metal electrodes, as indicated in Fig. 1a. An AFM image of the bare device 1 (Dev-1) with 2.4-nm-thick MoS$_2$ transferred onto Cr/Au electrodes is shown in Fig. 1c (before spin coating the P(VDF-TrFE) layer). Table 1 lists the parameters of MoS$_2$ devices studied in this work. Compared to the previous investigations with metal electrodes on the top of MoS$_2$ or other 2D materials[35,38,41], which may screen the FE polarization field near contact and result in Schottky barriers, this will ensure the intimate coupling of the FE polarization field to MoS$_2$ at the contacts and hence uniform doping to the whole semiconductor channel.

To switch the FE polarization in P(VDF-TrFE), an AFM system is employed for the facile reconfiguration of polarization pattern, as reported in the literature[22]. Despite the switched FE polarization tends to relax due to the incomplete compensation to depolarization field in P(VDF-TrFE), it enables rewritable polarization pattern on the same device, thereby allowing the direct study of the influence of device configurations without worrying about material differences. To pole the FE layer, the AFM tip is grounded while the source (S) and drain (D) electrodes of MoS$_2$ channel are biased by a poling voltage $V_p$. By scanning the AFM tip over the device area, the P(VDF-TrFE) copolymer between the tip and MoS$_2$ channel is polarized to P↑ using positive $V_p$ and P↓ with negative $V_p$. The effect of resulted remnant FE polarization in P(VDF-TrFE) to the electrical conductance of MoS$_2$ in Dev-1 is reflected by the significant hysteresis shown in Fig. 1d, which is measured with $V_{ds} = 1$ V after each polarization scan. With positive $V_p$, the current flow in MoS$_2$ is dramatically decreased with increasing $V_p$ due to the depletion of electrons. However, an apparent change is observed with $V_p > +20$ V, that is, increasing the poling voltage leads to increased conductance. Such transition is consistent with a reversal of the doping polarity in MoS$_2$ to p-type due to the dramatically increased remnant polarization (P↑) in FE film[16]. This is confirmed by the field-effect modulation measurements using the Si back-gate, as will be discussed later. By wweeping the

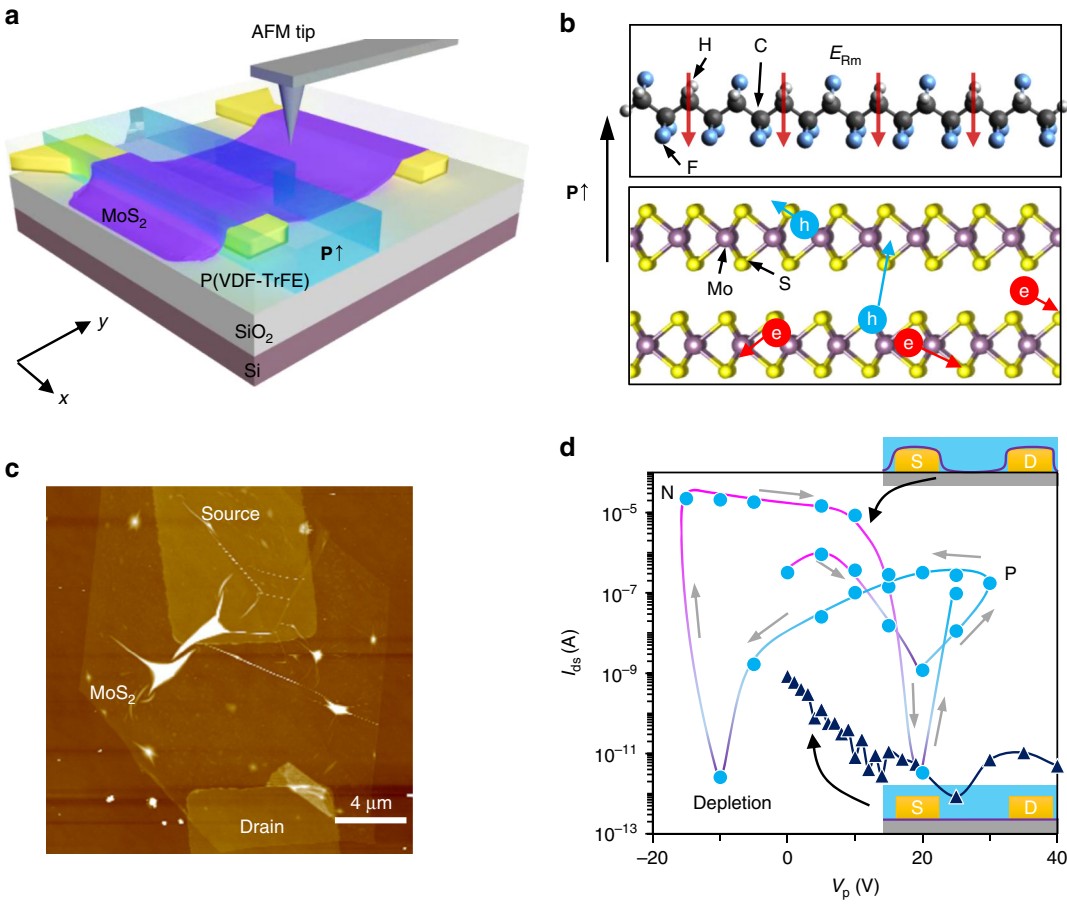

**Fig. 1** Ferroelectric polarization enabled p and n doping switch of MoS₂. **a** Configuration of ferroelectric copolymer-coupled MoS₂ device using a scanning atomic force microscope (AFM) tip as the poling electrode, the blue colored area becomes upward polarized (**P↑**) after applying positive bias on device while keeping the scanning AFM tip as ground. **b** The coupling between **P↑** polarization in ferroelectric copolymer and carriers in MoS₂ resulting in accumulated holes near the interface. **c** AFM image of a MoS₂ thin flake in device 1 (Dev-1) transferred onto predefined source and drain electrodes made by 10 nm Cr/Au. **d** Hysteresis variation of the MoS₂ conductance when sweeping the poling voltages ($V_p$) under different device configurations: with the electrode contacts defined at the bottom (circle) and on the top (triangle) of MoS₂, respectively

**Table 1 Parameters of MoS₂ devices and their defined functions in this work**

| Device | Length (μm) | Width (μm) | Thickness (nm) | Defined functions |
|---|---|---|---|---|
| Dev-1 | 8.0 | 7.8 | 2.4 | p, n, pn diode |
| Dev-2 | 7.6 | 3.2 | 3.8 | pn diode, npn transistor |
| Dev-3 | 6.9 | 4.7 | 3.4 | pn diode, npn transistor |
| Dev-4 | 7.5 | 3.0 | 4.4 | npn transistor |

poling voltage to negative, an opposite transition from p to n happens at $V_p < -10$V. These turning voltages are found consistent with the previous measured coercive voltages of P (VDF-TrFE) films, thus validating the correlation between the changed pn doping states in MoS₂ and the FE polarization switching. It should be noted that the successful doping polarity switching in MoS₂ is in sharp contrast with the one that employed electrode-on-top configuration, which in our control experiment displays only monotonous decrease of conductance with increasing $V_p$ (see Fig. 1d). This confirms the vital role of electrode contact configuration in ensuring the doping polarity engineering in MoS₂ and agrees well with the early conclusion that the doping in 2D semiconductors may be retarded by the Schottky barrier near metal contacts[42,43]. The universality of the above-adopted strategy is further demonstrated by its application in few-layer WSe₂ (4.2 nm) (Supplementary Fig. 2), which manifests nearly

symmetric p/n doping transition at $V_p = \pm 6$ V because of its bipolar characteristic.

With the present re-designed electrode configuration, the conductance of MoS₂ channel can be modulated with an on–off ratio of $> 10^7$ as shown in Fig. 1d. Their corresponding current–voltage (I–V) characteristics can be found in Supplementary Fig. 3. This modulation ratio is notably 2–3 orders higher than any of the previous reports because the significantly enhanced ON current in the MoS₂ channels by avoiding the contact issue[12]. To get further insight into the resulting doping characteristics in MoS₂ by the coupled FE polarization, we analyze the field-effect measurements using the Si back-gate. Figure 2a displays the transfer curves of the MoS₂ back-gate transistor after poling operations at $V_p = 0$, −15, and +25 V. The initial MoS₂ is seen n doped, that is, increasing $V_g$ at back-gate enhanced its conductance, while after −15 V downward

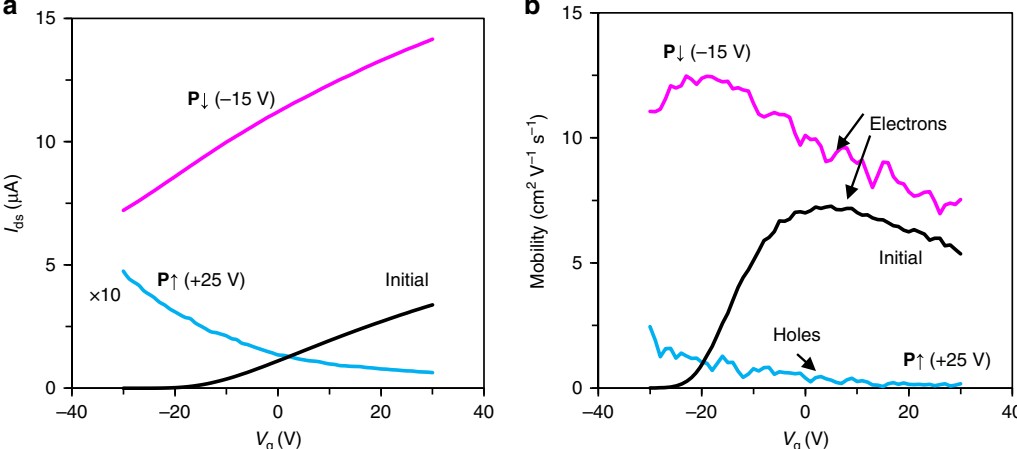

**Fig. 2** Field-effect characteristic of p- and n-doped MoS$_2$ under different ferroelectric polarization states. **a** Transfer curves of the MoS$_2$ device 1 (Dev-1) modulated by Si back-gate measured at a constant source–drain bias $V_{ds} = 1$ V. The device is measured in several polarization states including the initial state and after upward (**P↑**, by a poling voltage $V_p = +25$ V) and downward (**P↓**, by $V_p = -15$ V) ferroelectric polarization. **b** The corresponding mobility for electrons and holes extracted from field-effect measurements. The decrease of mobility when increasing gate bias for electrons is related to the electron–electron scattering limited mobility in degenerately n-doped MoS$_2$

polarization (**P↓**), its threshold voltage is remarkably shifted to negative from −13 to −85 V. Considering the oxide capacitance of 300 nm SiO$_2$, $C_{ox} = 11.5$ nF cm$^{-2}$, the FE-coupled effective charge carriers in MoS$_2$ is estimated to be ~0.8 μC cm$^{-2}$ by $\Delta Q = \Delta V_{th} \cdot C_{ox}$[44], which accounts to only a portion (~10%) of the remnant polarization of P(VDF-TrFE). This indicates that a majority of coupled electrons were frozen in new deep energy traps formed after polarization, for example, the acceptor states by the prompt interaction between aligned F-containing groups and MoS$_2$[45]. However, the mobility in MoS$_2$ is not influenced as the traps are effectively screened by the heavy concentration of electron carriers. Figure 2b displays the extracted carrier mobility in MoS$_2$ with respect to different FE polarization, calculated using $\mu = (L/W) V_{ds}^{-1} C_{ox}^{-1} (dI_{ds}/dV_g)$, where $L$ and $W$ are, respectively, the channel length and width. The maximum electron mobility in MoS$_2$ is ~12.3 cm$^2$ V$^{-1}$ s$^{-1}$ under **P↓**. Notably, increasing $V_g$ in positive leads to the decrease of electron mobility in n-doped MoS$_2$ due to the dominant electron–electron scattering, implying that the FE polarization-enhanced n-type doping in MoS$_2$ is reaching a heavily doped metallic behavior[6]. In comparison, the MoS$_2$ after 25 V (**P↑**) poling manifests apparent p-type behavior, with the hole mobility decreasing with increasing $V_g$. Such transport behavior is attributed to the trap-dominated transport in p-type MoS$_2$, since the inherent donor atoms in MoS$_2$ start to act as deep energy traps to holes[46]. By using the carrier mobility extracted from transfer curves at each FE polarization state, the free carrier concentration tuned by FE polarization is estimated to be ~10$^9$–10$^{12}$ cm$^{-2}$ in MoS$_2$ for both electrons and holes, and in WSe$_2$ ~10$^7$–10$^{11}$ cm$^{-2}$ (Supplementary Note 1, Supplementary Fig. 4, and Supplementary Table 1). The reversibly and significantly tuned p/n doping and large ON/OFF switch ratio covering metallic, semiconductor, and insulator behaviors will promote their potential applications in various optoelectronic devices with reprogrammable functions.

**Pn diode by locally defined FE polarization pattern**. Based on the FE polarization-enabled p and n doping in MoS$_2$, we construct a lateral pn junction using locally patterned polarization as illustrated in Fig. 3a. The bias voltage on the MoS$_2$ channel in experiment is programmed to switch from +25 to −15 V, while the tip scans between the S and D electrode, resulting in a polarization pattern shown in Fig. 3b, depicted by the phase

image in piezoelectric force microscope (PFM). Note that the expected transition region of remnant polarization (**P↑** to **P↓**) by switching the polarity of bias voltage is estimated to be ~100–180 nm (Supplementary Fig. 5, limited by the electric field distribution near AFM tip and the FE domain size in P(VDF-TrFE) at a thickness of 200 nm). Figure 3c displays the $I$–$V$ characteristic of MoS$_2$ channel in Dev-2 with three kinds of FE field-effect doping configurations, that is, complete p and n doping on the device area, and patterned lateral pn junction. For both the n- and p-doped device, the electrical contact shows Ohmic behavior due to the degenerate doping in MoS$_2$. However, we note that for moderately p-doped MoS$_2$, nonlinear $I$–$V$ characteristics may appear with ohmic contact at low bias conditions, but space charge-limited current (SCLC) at large biases, as revealed in Supplementary Fig. 3. Such behavior is however attributed to the charge trapping in channel upon intense hole injection under large external bias[47]. Nevertheless, SCLC behavior is alleviated in heavily p-doped devices by the filling of trap centers when the Fermi energy $E_F$ is close to the valance band maximum. With laterally patterned p and n doping of the channel, the formed pn diode exhibits clear rectification behavior with a large on–off ratio of ~10$^5$ at ± 1 V. The value is higher than the MoS$_2$/GaTe heterojunction[48] and larger than the lateral MoS$_2$ homojunction enabled by chemical doping[49], which are attributed to the widely tuned doping concentration in MoS$_2$ together with the electrical contact. By fitting the junction current using

$$I = I_0 e^{e(V - IR_s)/nk_B T}, \qquad (1)$$

where $I_0$, $R_s$, and $n$ denotes, respectively, the reverse saturation current, series resistance, and ideal factor. The formed pn junction is found to exhibit an ideal factor of 1.7 and $R_s \approx 10$ MΩ. With $n$ ~2, the junction current shall be dominated by electron–hole recombination in the space charge region[50]. The recombination could however be optimized by, for example, improving the collection efficiency of metal electrodes by minimizing the size of the p region. In our work, the ideal factor is reduced to 1.4 when reducing the p-doped region in junction (Supplementary Note 2, Supplementary Fig. 6–8), which implies that within the FE-defined pn diode the majority of the charge recombination occurred at the p-doped region.

When illuminating the pn diode with 532 nm light, apparent self-driven photocurrent is observed under zero bias driven by the

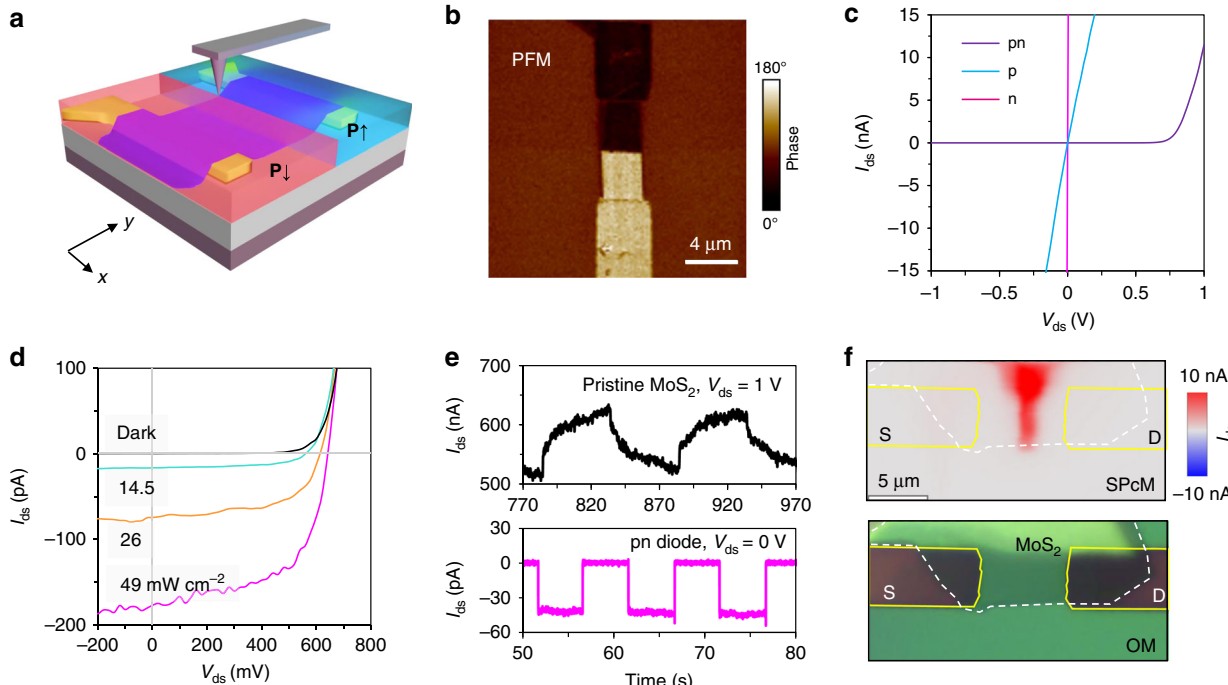

**Fig. 3** MoS$_2$ pn junction defined by locally patterned ferroelectric polarization and its characteristics. **a** Formation principle of MoS$_2$ pn junction by locally patterned upward (**P↑**) and downward (**P↓**) ferroelectric polarization using a scanning AFM tip. **b** Piezoelectric force microscope (PFM) phase image of the formed FE polarization pattern on device 2 (Dev-2). The bright and dark-colored region correspond, respectively, to the polarization states of **P↑** and **P↓** in FE copolymer, which cause, respectively, p- and n-type doping to MoS$_2$. **c** I–V characteristic of the pn-doped MoS$_2$ channel compared to complete p and n doping. **d** I–V characteristic of the MoS$_2$ pn junction under 532 nm laser illumination of different intensities, from dark to 14.5, 26, and 49 mW cm$^{-2}$, showing photovoltaic-like behavior. **e** Transient photoresponse of the self-powered pn junction compared to the pristine MoS$_2$ device in photoconductor configuration. **f** Spatial photocurrent map of Dev-3 in short circuit compared to its optical microscope image. Self-driven photocurrent is confirmed to generate from the pn junction defined in the middle of channel rather than from contact effects

built-in field in the junction as can be seen in Fig. 3d. In comparison, devices with complete n or p doping across the channel do not display such self-driven photocurrent under illumination (Supplementary Fig. 9). The open-circuit voltage ($V_{oc}$) of the FE polarization-defined pn junction is ~650 mV with light intensity >10 mW cm$^{-2}$, which is comparable to silicon diodes and approximately half of the indirect bandgap (1.2 eV) of few-layer MoS$_2$. The large $V_{oc}$ suggests efficient charge separation in devices enabled by the intense lateral built-in electric field across the pn diode. In the short-circuit condition, the photocurrent is ~177pA at an intensity of 49 mW cm$^{-2}$, indicating a self-powered responsivity $R \approx 15$ mA W$^{-1}$ ($R = \alpha\eta$ ($e/hv$), where $\alpha$ and $\eta$ are, respectively, the light absorption efficiency and internal quantum efficiency in device). Considering that 3.8 nm MoS$_2$ absorbs ~30% of the incident photons at 532 nm[51], $\eta$ is estimated to be 12% in the diode, which could be optimized by defining the junction close to electric contact, so as to minimize the serial resistance in the channel. Given the large $V_{oc}$ and a fill factor of ~0.58, an overall power conversation efficiency of ~0.61% is achieved. We note that the value outperforms that of vertically stacked GaTe/MoS$_2$ heterojunction[48], MoS$_2$ pn junctions[52], and the lateral WSe$_2$/MoS$_2$ pn heterojunctions[53], suggesting the attractive potential of FE-coupled pn junctions. As indicated in Fig. 3e, the efficient photovoltaic separation of electrons and holes renders rapid self-powered photoresponse in photodetection over conventional photoconductor devices. The self-powered MoS$_2$ device exhibits fast response within 10–20 μs (Supplementary Fig. 10), being ~6 orders of magnitude faster than the pristine MoS$_2$ that suffered persistent photoconductance by the long-lasting photogate effects from trap states[54].

To elucidate the self-powered photocurrent generation, spatially resolved photocurrent distribution has been measured for a short-circuited pn diode defined in Dev-3. This is achieved by locally illuminating the device using a fine laser spot ($\lambda = 532$ nm, focused spot diameter ≈500 nm) in a confocal microscope. The pn junction is defined in the middle of MoS$_2$ channel. Figure 3f displays the optical microscope image of the device and the associated photocurrent map. It is clear that most of the photocurrent is generated near the defined junction. We note that the self-driven photocurrent may also appear in Schottky-contacted devices, but usually with reversed polarity near the source and drain electrodes due to opposite charge separation[55–57]. However, in all the devices we have studied, photocurrent barely appears near the contact electrodes and there is no change on the photocurrent polarity across the device area. These results thus validates the role of FE polarization-defined pn diode in bringing the high rectification in I–V characteristics and self-driven photocurrent.

**High gain npn bipolar transistor by FE polarization**. Although pn diodes exhibit remarkably improved photoresponse speed over photoconductive devices, their responsivity is limited due to the loss of gain mechanism[58]. Here, with pn junction as the fundamental building block, we further construct a bipolar phototransistor in which the fast photovoltaic photocurrent can be amplified via its transistor action[34]. Compared to the avalanche photodetectors (APDs), the bipolar phototransistor could work at considerably lower operation voltages (~150 V for commercial Si APDs) while yielding the similar photodetection gain. A typical bipolar transistor consists of emitter (E), base (B), and collector (C) regions that are, respectively, n, p, and n doped by FE

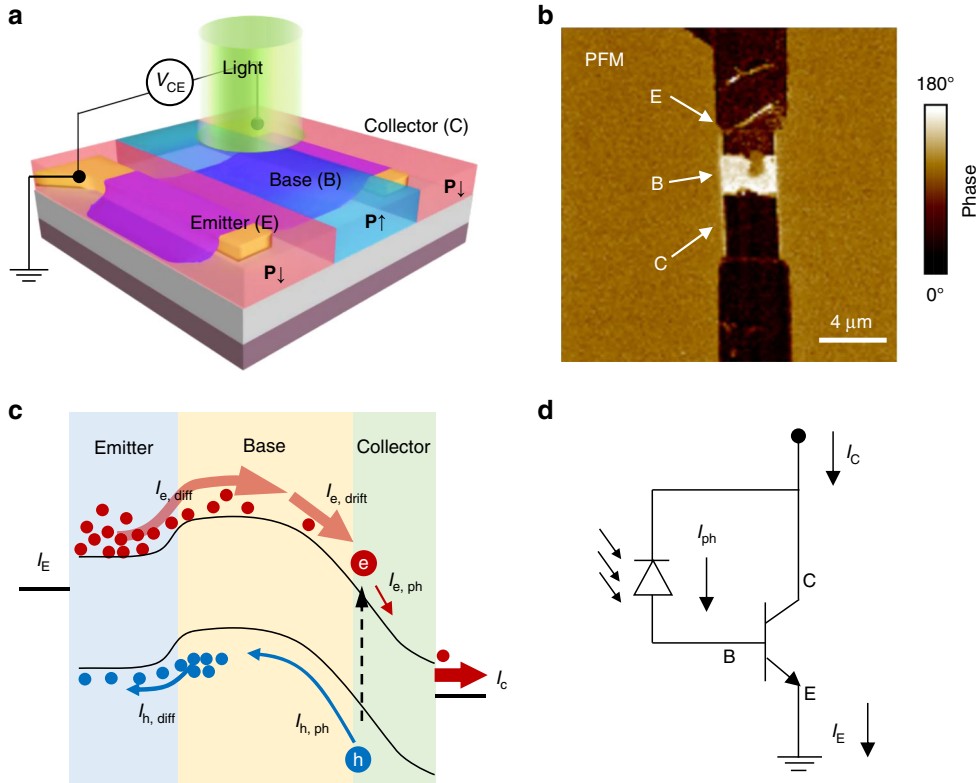

**Fig. 4** MoS$_2$ bipolar phototransistor defined by ferroelectric polarization. **a** Schematic illustration of the fabrication of npn phototransistor by laterally patterned ferroelectric polarization (**P↓**, **P↑**, and **P↓**) across the device, forming, respectively, the emitter (E), base (B), and collector (C) of the bipolar transistor. **b** Piezoelectric force microscope (PFM) phase image of the ferroelectric polarization pattern in device 2 (Dev-2) to make a MoS$_2$ bipolar transistor. **c** The energy band diagram and **d** equivalent circuit illustration of the working principle of a npn phototransistor under bias. The photovoltaic current ($I_{ph} = I_{e,\,ph} + I_{h,\,ph}$) at the reverse biased base–collector (B–C) junction is amplified by the large electron injection flux ($I_{e,\,diff}$) at the forward biased emitter to base junction (E–B), which is swept to collector when reaching the reverse biased B–C junction. The collector current $I_C$ is finally amplified from photovoltaic current $I_{ph}$ by a factor of $\beta \approx I_{e,diff}/I_{h,diff}$ related to the electron and hole doping concentration in the emitter and base

polarization, as illustrated in Fig. 4a. The PFM phase image of the polarization pattern defined on Dev-2 for a npn bipolar transistor is displayed in Fig. 4b. For photodetection, the as-formed bipolar transistor is operated in a common-emitter mode with the base floated and collector positively biased. Since the transistor has the structure of two reversely connected pn junctions, it exhibits low dark current similar to that of a reversely biased photodiode. However, substantial gain could be obtained in the bipolar transistor to the photovoltaic current at the reverse biased B–C junction[28]. This is elucidated by the energy band diagram in Fig. 4c, where E–B junction is forward biased and B–C junction is reverse biased under the positive collector bias. Upon illumination, the photogenerated electrons and holes within the B–C junction are separated with electrons drifting to the n-type collector while holes drift to the p-type base, thus giving rise to the photocurrent $I_{ph}$ as that in the photodiode. The holes separated into the base then diffuse across the forward biased E–B junction as $I_{h,diff}$, and induce a significant diffusion flux ($I_{e,diff}$) of electrons in the opposite direction. Once the electrons diffuse across the base and reach the reverse biased B–C junction, they will be swept and collected to the collector terminal by the built-in electric field, therefore contributing to the overall collector current. The collector current $I_C$ is then amplified from the initial photocurrent $I_{ph}$ by the injected electrons from E. With $I_C = (1 + \beta) \cdot I_{ph}$, $\beta$ is known as the gain factor that qualifies the performance of bipolar transistor, which is generally determined by the electron injection efficiency over the reverse hole current at the forward biased E–B junction by $\beta \approx I_{e,diff}/I_{h,diff}$. Since the diffusion current across the

forward biased junction is governed by the majority carrier density over the space charge region, $\beta$ is intimately related to the doping concentration in the E and B with $\beta \propto N_e/P_b$, where $N_e$ and $P_b$ are, respectively, the electron and hole density in E and B[59]. The operation principle of such bipolar phototransistor is illustrated by the equivalent circuit shown in Fig. 4d, in which the diode corresponds to the reversely biased B–C junction and the transistor represents the amplification of photodiode current by the forward biased E–B junction. In the present case, due to doping compensation effect in MoS$_2$ by the inherent donor atoms, hole doping by FE coupling is naturally less significant than the case of electrons. A substantial gain factor to the photocurrent could be in principle obtained by constructing a npn transistor based on FE polarization.

Figure 5a shows the photoresponse behavior of the as-formed npn bipolar transistor to 532 nm laser illumination with varied intensity ranging from 0.02 to 39 mW cm$^{-2}$. Once configured into bipolar transistor, the photocurrent is found remarkably improved in comparison to the previous photodiode along with a fast response speed. Such improvement is consistent with the expected behavior of bipolar transistor that amplifies photocurrent with high gain. By using a fast-switching 365-nm LED source, the photoresponse speed of the bipolar transistor is estimated to be ~20 µs (see inset of Fig. 5a), making it one of the fastest MoS$_2$ photodetectors, but with high gain characteristics. Faster response within as short as ~3–5µs is also achieved in experiments (Supplementary Fig. 11), which is close to the switching limit of the adopted light source. It is believed that the

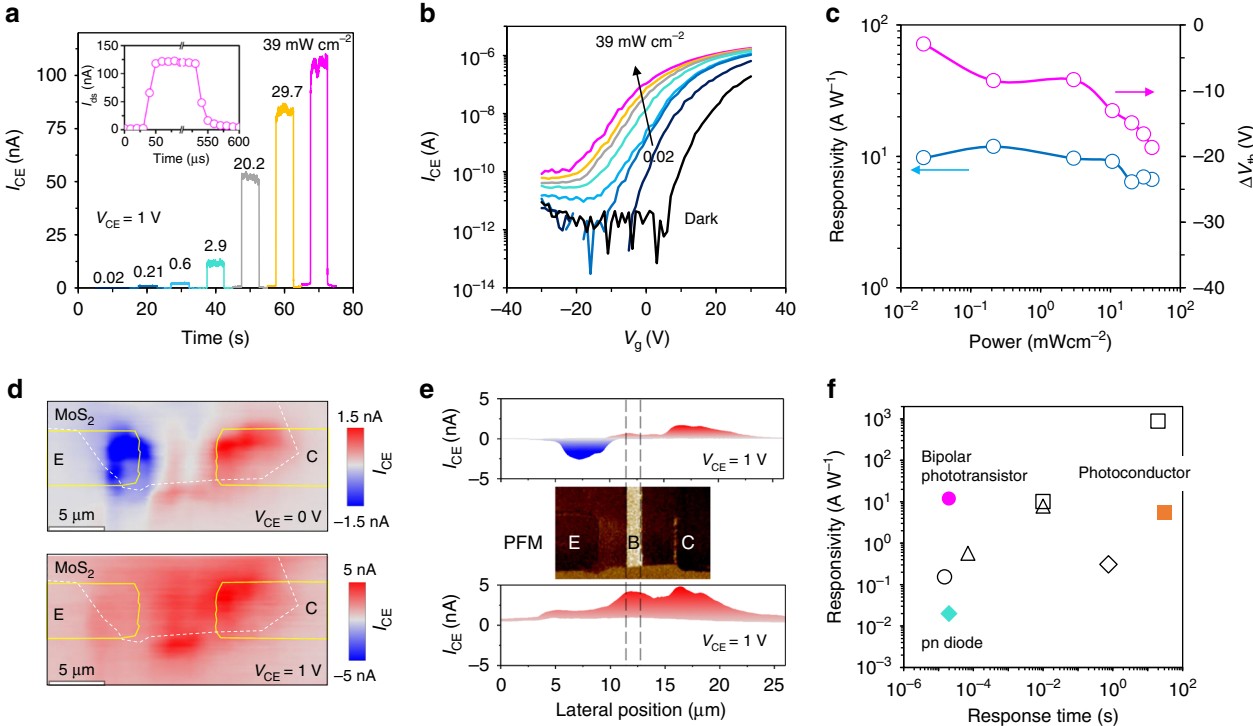

**Fig. 5** Photodetection performance of MoS$_2$ bipolar phototransistor. **a** Photoresponse of ferroelectric polarization-coupled MoS$_2$ npn transistor (device 2 (Dev-2)) under varied laser illumination intensity from 0.02 to 39 mW cm$^{-2}$, and **b** the corresponding transfer curves measured using Si back-gate under a constant collector to emitter bias $V_{CE} = 1$ V. **c** Responsivity and the extracted threshold voltage shift $\Delta V_{th}$ of the transistor at varied light illumination intensity. **d** Spatial photocurrent map of the npn bipolar phototransistor (Dev-3) under $V_{CE} = 0$ and 1 V. **e** compares the lateral distribution of photocurrent to the PFM phase pattern used to define the npn transistor. The reversed photocurrent polarity at $V_{CE} = 0$ V near emitter (E) and collector (C) terminals originates from the reverse pn junctions of emitter to base (E–B) and base to collector (B–C). Under $V_{CE}$ bias, most photocurrent is generated near the reversely biased base–collector (B–C) junction for the intense electric field there. **f** Comparison of the photodetection performance of MoS$_2$-based devices (filled symbols, this work and open symbols, literature: refs. [3,18,34,63–65]), including photoconductors (square), photodiodes (diamond), bipolar phototransistors (circle), and other types of phototransistors (triangle)

ultimate device response speed depends on both material characteristics and device geometries. Further improved speed is likely attainable given higher carrier mobility in MoS$_2$ and improved design on the width of base and collector, as they directly determine the overall carrier transit time in device. In Fig. 5b, the phototransistor behavior is further characterized by the transfer curves obtained by the back-gate field-effect modulation. As the current in the npn junction is limited by the generation rate of the minority electrons in the base, the as-formed bipolar transistor manifests n-type conductance that increases with increasing $V_g$ during field-effect measurements. When shining light on the device, the threshold voltage $V_{th}$ is apparently negatively shifted, implying an increase of electron concentration in the base by the forward injection from the emitter. The shift of threshold voltage is consolidated in another npn bipolar transistor defined on 8-nm-thick MoS$_2$ device (Supplementary Fig. 12). The responsivity of the npn transistor is extracted and compared to the shift of $V_{th}$ under varied illumination intensity, as shown in Fig. 5c. $R$ closely follows the variation of $V_{th}$, implying the important role of injected electrons in amplifying the photoresponse. $R$ slightly decreases with the negative shift of $V_{th}$ at higher light intensity, which is also usually found in other type phototransistors exhibiting a photogate effect. However, we emphasize that the origin of such dependence is different from the usual saturated charge trapping or separation in phototransistor, but due to the increasing recombination losses at the forward biased E–B junction under large injection[28]. In Fig. 5d, e, we present the photocurrent map for the present npn bipolar phototransistor under $V_{CE} = 0$ and 1 V to validate its

operation principle. It is seen that photocurrent of reversed polarity appears near C and E terminal at $V_{CE} = 0$ V, while at $V_{CE} = 1$ V the photocurrent is more efficiently generated near the reversely biased B–C junction. This is consistent with the expected electric field strength in a device that eventually separates the photogenerated electron–hole pairs. The present photocurrent map also differs from other type of phototransistors that usually displays uniform photocurrent distribution within the photogate area[60–62]. Such difference in the two kinds of phototransistors is ascribed to their different gain generation mechanism, that is, via the lateral in-plane charge injection and the out-of-plane photovoltaic effects, respectively.

By virtue of the low dark current, the detector could be optimally operated without the assistant of external gate biases, yielding $R$ ~12 A W$^{-1}$ at 0.2 mW cm$^{-2}$ and high shot-noise-limited detectivity of 10$^{13}$ Jones according to $D^* = RA^{-1/2}/(2eI_{dark})^{-1/2}$, where $A$ is the device area. As a bipolar phototransistor, the device is found to exhibit 10 times higher responsivity over the initial photoconductor while preserving a fast response speed similar to the photodiode (see Fig. 5e). Compared to the photodiode, a substantial gain factor of ~1000 is expected from the defined bipolar transistor to yield such an improvement to responsivity. The value is however substantially larger than any previous reported bipolar transistors based on buried gate[28], lateral heterojunctions[29], or vertically stacked heterojunctions[34]. As indicated in Fig. 5f, when compared to other MoS$_2$ photodetectors in either photoconductors[3,18], photo-diode[63], or the phototransistor configurations[34,64,65], the high gain value gives rise to competitive photodetection performance

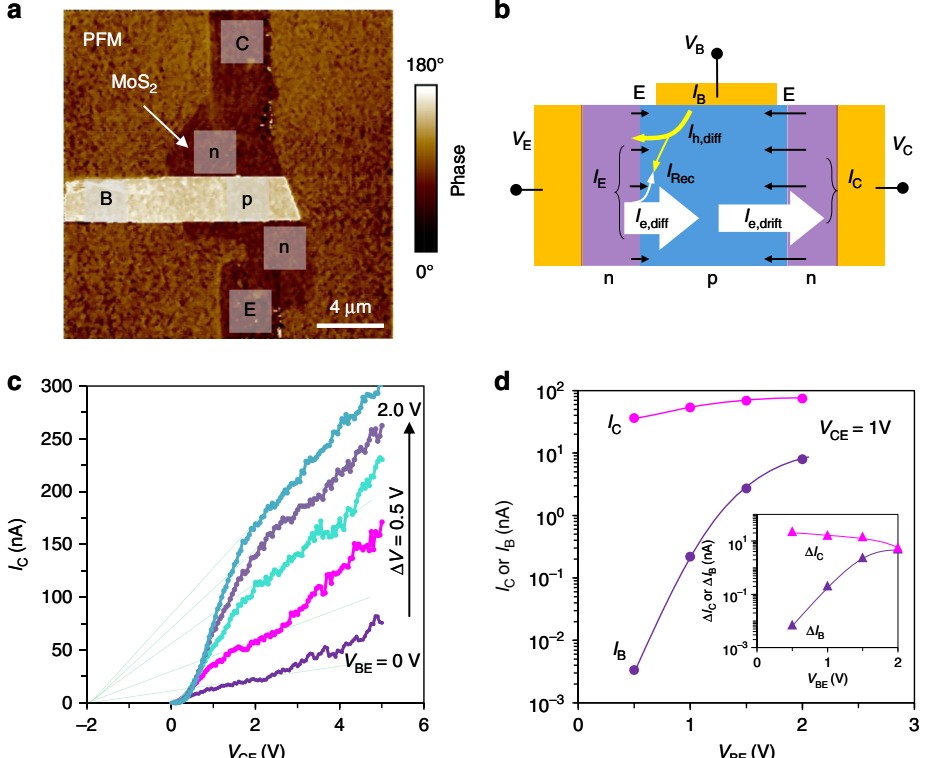

**Fig. 6** Ferroelectric polarization-patterned three-terminal $MoS_2$ npn bipolar transistor and its characteristic. **a** Piezoelectric force microscope (PFM) phase image of a three-terminal bipolar transistor with base connected to external terminal. **b** Illustration of the current components that contribute to the overall collector current $I_C$. The input base current $I_B$ (contributed mainly by hole diffusion, $I_{h, diff}$) is amplified by the triggered electron diffusion current at the emitter ($I_{e, diff}$), which is then swept to the collector terminal ($I_{e, drift}$) by the intense electric field (indicated by black arrows) at reversely biased junction. Recombination loss ($I_{Rec}$) during diffusion through base shall be suppressed to maximize the amplification ratio. **c** Output characteristic of the three-terminal bipolar transistor under varied base input voltages $V_{BE}$. Increasing $V_{CE}$ at a constant $V_{BE}$ results in larger $I_C$ due to the extended space charge region in reverse biased base–collector (B–C) junction and due to the reduction of effective base width, which promotes electron diffusion across the base. Extrapolation to $I_C = 0$ under different $V_{BE}$ yields a consistent early voltage of approximately −2 V. **d** Dependence of the collector $I_C$ and base current $I_B$ to $V_{BE}$ input. The inset displays the extracted $\Delta I_C$ and $\Delta I_B$ at each $V_{BE}$ step, from what a substantial gain ($\beta = \Delta I_C / \Delta I_B$) ≈3000 is derived for the transistor action

by delivering simultaneously fast speed response speed and high responsivity (see detailed comparison in Supplementary Table 2).

In order to confirm the high performance of the bipolar transistor, we fabricate a three-terminal device by using $MoS_2$ of similar thickness ~5 nm. Figure 6a displays the PFM phase pattern of the FE polarization for the device. The base terminal is connected to metal electrode contact for a direct input of the base current ($I_B$), thus to probe the amplification at collector end. The current that contributed to the overall output at $V_C > V_B > V_E$ is schematically illustrated in Fig. 6b. The output characteristic of the transistor is shown in Fig. 6c, which is measured by sweeping $V_{CE}$ under different $V_B$ input from 0 to 2 V and in the presence of the background laser of AFM feedback system. Here, the collector current $I_C$ is significantly modulated by the input $V_B$. Note that $I_{CE}$ increased with increasing $V_{CE}$ rather than being a constant due to the modulation to the base width by the extended space charge region at reverse biased B–C junction[59]. Such base width modulation results in a small early voltage of ~2 V (the extrapolated intersection with $V_{CE}$), which is related to the low doping concentration in base. Slight increase of $V_{CE}$ would dramatically modulate the effective base width and the diffusion current across the base. From the measured output characteristic, the amplification behavior of the bipolar transistor is extracted at $V_{CE} = 1$ V and shown in Fig. 6d. The collector current $I_C$ scaled with increasing $V_{BE}$, and is 2–3 orders of magnitude larger than $I_B$. The gain factor observed in photodetection could be probed

from the magnitude of $I_C$ variation while changing $I_B$. As shown in the inset of Fig. 6d, when changing $V_{BE}$ from 0 to 1 V, we obtain an superior gain factor $\beta$ ~3000, which is on the same order of magnitude as the expected amplification ratio in bipolar phototransistor as discussed above. The gain factor $\beta$ decreases gradually to 1 when $V_{BE}$ is increased, following closely the case in photodetection that responsivity reduces at higher light illumination intensity. Interestingly, at $V_B = 1.5$ V, with both B–C and B–E junction forward biased, one still obtains $\beta$ ~10. This clearly suggests the significant electron diffusion flux from emitter to collector in contrast to the hole flux in the same direction at the forward biased base to collector junction[29]. Our results therefore unambiguously confirm the high gain characteristic of FE-coupled bipolar phototransistor, which originates from the widely and fine-tuned doping states in $MoS_2$.

## Conclusions

In summary, we have demonstrated the facile reconfigurable customization of $MoS_2$ optoelectronic devices using the rigid pn doping engineering enabled by switchable FE polarization. By polarizing the $MoS_2$ device into homojunctions of pn diode and npn bipolar transistors, the device is configured into self-powered or high gain photodetectors of optimal performances without the assistance of external gate bias. The gate-free yet reconfigurable methodology introduced the great potential of exploiting locally

coupled FE polarization in customizing high-performance optoelectronic devices based on the thriving 2D semiconductors and in the future their van der Waals heterojunctions, which in principle could offer even larger speed and gain product than present homojunctions. Further maturation of such strategy towards array-structured functional optoelectronic devices with high stability shall be viable based on predefined top-gate patterns or alternatively flexible electrical imprint methods.

## Methods

**Device fabrication**. Few-layer $MoS_2$ with thickness ranging from 2 to 5 nm are exfoliated from $MoS_2$ single crystals (Nanjing MKNANO Tech. Co., Ltd.) using scotch tape and transferred to $SiO_2$/Si substrate by polydimethylsiloxane stamp. The electrode contacts are made by Cr/Au electrode. In the case of bottom electrode contacts, Cr/Au (5/5 nm) electrodes are first defined on the substrate using lithography processes with direct laser writing. The exfoliated $MoS_2$ thin flakes are later transferred. The as-prepared devices are then annealed at 150 °C with Ar protection for 1 h to release stresses and improve electrical contact. The FE thin film of P(VDF-TrFE) copolymer is then spin coated on the device using a solution of 2–5 wt% P(VDF-TrFE) (Piezotech FC 25) dispersed in anhydrous N,N-dimethylformamide (99.8%, Alfa Aesar). The copolymer film is then annealed at 135 °C for 15 min to improve the crystallinity.

**Local FE polarization**. An AFM system (Dimension Icon, Bruker) is used to realize the FE poling using scanning conductive AFM tip with the $MoS_2$ source and drain electrode biased by a poling voltage $V_p$. A potential bias of 0.1 V is kept between source and drain for the in situ observation of the $MoS_2$ conductance during the polarization scan. To obtain local polarization coupling to $MoS_2$, $V_p$ is programmed during the AFM scan.

**Device characterization**. The device after polarization is placed in a Lakeshore probe station (TTPX, Lakeshore) equipped with a semiconductor device analyzer (B1500A, Agilent) for electrical measurement. After AFM poling, the FE polarization in device tends to degrade within 1–2 h because of the lack of screening to depolarization field without top metal contacts. The device performance is therefore studied within this period; out of the period the device is repolarized at the same conditions for measurements. A 532-nm laser with tansistor-trasistor logic triggering is used as the light source for photodetection measurements, in which a power meter (FieldMaxII-TO, Coherent) is used to calibrate the light intensity. To probe the photoresponse speed, a 365 nm fast-switching LED (M365FP1, Thorlabs) and fast measurement unit (B1530, Agilent) is used.

## Data availability
The data that support the plots within this paper and other findings of this study are available from the corresponding authors upon reasonable request.

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

## Acknowledgements

This work was supported by National Natural Science Foundation of China (Grant Nos. 61804059, 21825103, and 51727809), National Key Research and Development Program of "Strategic Advanced Electronic Materials" (Grant No. 2016YFB0401100), and the Fundamental Research Funds for the Central University (Grant No. 2019kfyXMBZ018). F.W.Z. owes special thanks to Dr. M.M. Yang at the University of Warwick for the discussion.

## Author contributions

F.W.Z. and T.Y.Z. conceived the idea. L.L., F.J.X., X.J.X., and N.Z. prepared and characterized P(VDF-TrFE) polymers, MoS$_2$, WSe$_2$ flakes, and the devices. L.L., Q.F.Z., and Y.H. performed PFM polarization, device characterization, and photocurrent mapping. F. W.Z., T.Y.Z., and L.L. analyzed the data and wrote the manuscript together with discussion with all authors.

## Additional information

**Competing interests:** The authors declare no competing interests.

