## [Peer Review File · Nature Communications]

Reviewers' comments:

Reviewer #1 (Remarks to the Author):

The paper reports on the formation of lateral diodes in multilayered MoS₂ using different polarization states of the covering organic ferroelectric P(VDF-TrFE) copolymer. The ferroelectric nature of P(VDF-TrFE) in contact with MoS₂ was used to manipulate the carrier doping in MoS₂ based on their reversible polarization by external polling field provided by a biased AFM tip. The idea of using different polarization states to create a lateral diode is novel, and the experimental results of MoS₂ optoelectronic device are convincing, however, there are several issues that should be addressed before publishing in Nature Communications.

1. Why the authors did not use the monolayer WSe₂ in their optoelectronic device?
2. Can the p- and n-doping concentrations of WSe₂ be finetuned by modulating the amount of polarization induced in the ferroelectric copolymer? Since authors claimed that "early attempts to construct the kind of devices relied prominently on local-buried gates, lateral and vertical heterojunctions, the behavior of which was complicated to manipulate without finetuned pn doping." (line 49-52 on page2) If yes, what are the ranges of the p- and n-doping concentration tuning?
3. The following question: The P-V hysteresis is very symmetrical (Supplementary Fig. 1). Can the authors estimate the transition region of doping from P_{up} to P_{down}?
4. The authors claimed that "FE coupling to 2D semiconductors is favored to realize reconfigurable optoelectronics based on rewritable pn doping in the 2D components, which however hasn't been demonstrated yet." (line 57-59 on page3) However, in this manuscript, all MoS₂ devices are not made by the same sample, this means that the authors do not demonstrate the rewritable capability in their configuration either. Can the authors comment on this?
5. About the rectification shown in Fig. 3b, asymmetric I-V characteristics are frequently seen in 2D materials such as WSe₂, MoS₂, etc, due to the Schottky barriers at the contacts. Could the authors comment on how the effects of the contacts are accounted for?
6. I am also missing a justification for the current-voltage characteristics found in Supplementary Fig. 2. If that was for gate modulated current in pn diodes at the dark and illuminated condition for different configurations to get qualitative comparisons, it should be mentioned in the paper.

Reviewer #2 (Remarks to the Author):

The authors present lateral p-n and n-p-n junctions based on a two-dimensional semiconductor (MoS₂), enabled by local patterning of the ferroelectric polarization in P(VDF-TrFE). Using this technique, they realize reconfigurable devices, such as diodes and bipolar transistors, and demonstrate optoelectronic applications such as photovoltaic energy conversion and photodetection.

Although the idea of using ferroelectrics for such purpose is not entirely new, this work constitutes an important advance in this emerging field as the device performance appears to be dominated by the properties of the junctions, rather than the contacts. I thus would like to recommend publication of the manuscript in Nature Communications, provided that the authors can satisfactorily address the following comments/issues:

- In Fig. 3c the authors present photovoltaic properties of their p-n junction. (It is not clear how the data were taken; I assume the whole device was illuminated?) The authors should present the same measurements for n-n and/or p-p configuration to exclude the possibility that the photoresponse stems from the (possibly asymmetric) metal/MoS₂ Schottky junctions. Alternatively, the authors may choose to locally illuminate the p-n junction.
- Device operation as bipolar transistor (Fig. 6) is convincingly demonstrated. When operated as phototransistor, though, the device shows some behaviors that resemble those commonly seen in pristine MoS₂ devices: namely, the drop of the photoresponsivity with illumination intensity and

the rather slow response time (4 ms). The authors argue by comparison with a pristine MoS₂ device (Fig. 3d) that the response times observed in their junctions are shorter, but 4 ms still seems very long to me. Can the authors provide further evidence that the gain stems from the transistor operation, rather than charge trapping in (short-lived) defects?

- It would also be helpful if the authors could comment on the stability of the junctions (versus time, bias voltage, etc.)

Reviewer #3 (Remarks to the Author):

In this work, the authors report some interesting results on the integration of MoS₂ with ferroelectric materials. The authors demonstrated reconfigurable photodetectors with high response. Overall the results are interesting but I am not convinced completely that this work should be published in Nature Communications.

(1) It is not clear why such reconfigurability is needed. We can easily apply an electric bias to tune the device operational condition. Moreover, there are many different approaches to make photodetectors and the introduction of ferroelectric materials into optoelectronic devices do not seem to be well-justified, at least in this case. You can make a pn junction or directly make a phototransistor. You can also make an APD. Sometimes you do need reconfigurable photonic devices (e.g. in optical networks) but I do not think here making a reconfigurable photodetector has intrinsic advantages.

(2) The performance of the photodetector is not very impressive. Indeed the responsivity is high due to the gain. However the response time is long (4 mS) and as a result, the speed is very low (below kHz). It is very easy to achieve high responsivity if you do not care about the speed. The difficult part is high responsivity, high speed and low noise simultaneously. In fact, such a high gain can be easily achieved in a simple silicon photoconductor if high speed is not needed.

Reply to reviewers' comments

Firstly, we would like to thank all the reviewer's precious comments, which helped us to improve the work dramatically.

The major changes to manuscript in this revision were briefly listed as following:

1. Figure 3, 4, 5 were updated using the pn photodiode and npn bipolar transistor configured on the same device, therefore addressing the reconfigurability in devices by rewritable FE polarization.
2. The photoresponse speed of pn diode and npn phototransistor was measured again using fast switching light source and fast measure unit. The results showed 10-20 μ s fast response of the devices, among the fastest in all kinds of 2D photodetectors.
3. Photocurrent map of the pn diode and npn transistors were added in Figure 3f and Figure 5d, to exclude the Schottky contact effect in rectification and clarify the operation principle of npn bipolar phototransistor.
4. Supplementary material was reorganized into section I-V for better understand. Fig. S2, S3, S4, S6, S10, Table S1 and S2 were added to address the concerns raised by reviewers. Figure S7, and S11 previously in main manuscript were now moved into supplementary material.
5. Table 1 was added to the manuscript to clearly present the device parameters in study.
6. A mistake was found in the calculated detectivity due to improper unit transformation. This was now corrected in this revision with an optimal dark current limited detectivity of 10^{13} Jones for npn phototransistor.
7. Some English and grammar mistakes have been corrected.

In the following pages, the reviewers' comments were replied point to point. Note that Reply to comments was marked in blue, while the related changes in manuscript were marked as purple.

Reviewer #1 (Remarks to the Author):

The paper reports on the formation of lateral diodes in multilayered MoS₂ using different polarization states of the covering organic ferroelectric P(VDF-TrFE) copolymer. The ferroelectric nature of P(VDF-TrFE) in contact with MoS₂ was used to manipulate the carrier doping in MoS₂ based on their reversible polarization by external polling field provided by a biased AFM tip. The idea of using different polarization states to create a lateral diode is novel, and the experimental results of MoS₂ optoelectronic device are convincing, however, there are several issues that should be addressed before publishing in Nature Communications.

We sincerely appreciate all the reviewer's comments that helped us to improve the manuscript.

1. Why the authors did not use the monolayer WSe₂ in their optoelectronic device?

Reply:

We agree with the reviewer that WSe₂ was a good choice for reversible p/n doping by ferroelectric polarization for its bipolar characteristics. Unfortunately, in our initial experiments, the exfoliated WSe₂ thin flakes exhibited poor contact properties when transferred onto electrodes. We expected this issue came from the fast oxidation of WSe₂ in ambient conditions given that selenides are less stable than sulfides (Li et al. J. Mater. Chem. A, 2019, DOI:10.1039/C8TA10306B). The difference in as-obtained crystal quality in terms of the anion vacancy may also contribute in such difference, since it facilitates the oxidation process by exposing metal elements on surface (reaction enthalpy for oxidation near anion defect of WSe₂: -2eV, MoS₂: -1.7eV, according to Dabral et al. Phys. Chem. Chem. Phys. 2019, 21, 1089).

To avoid the stability issue, we now spin-coated P(VDF-TrFE) layer right after transferring 2D materials onto electrodes. In this way, we were able to obtain reversible pn doping in WSe₂ (3 nm) as we have presented in MoS₂. As indicated in the new Figure S6 in revised supplementary materials, the ON/OFF switching ratio in WSe₂ was $>10^6$, and the initial bipolar characteristic of WSe₂ was tuned to P and N by a poling voltage $\sim\pm 6$ V. To demonstrate the universality of the method, we have added the polarization results on WSe₂ as Figure S3 in supplementary materials.

Supplementary Figure 3. FE polarization tuned conductance in MoS₂ and WSe₂ devices. (a), (d) AFM image of measured MoS₂ and WSe₂ device. (b), (e) FE polarization tuned conductance in MoS₂ and WSe₂ channel, and (c), (f) their corresponding transfer curves after each polarization state when sweeping V_p in the negative direction (from 14 to -8V in c, and 5 to -15V in f).

The following discussion is now included in Page 6 line 4-7 in the revised manuscript to reflect the change in Figure S6.

“The universality of present strategy was further demonstrated by its application in few layer WSe₂ (4.2 nm) (Supplementary material section II), which manifested nearly symmetric p/n doping transition at $V_p = \pm 6V$ because of its bipolar characteristic.”

2. Can the p- and n-doping concentrations of WSe₂ be finetuned by modulating the amount of polarization induced in the ferroelectric copolymer? Since authors claimed that “early attempts to construct the kind of devices relied prominently on local-buried gates, lateral and vertical heterojunctions, the behavior of which was complicated to manipulate without finetuned pn doping.” (line 49-52 on page2) If yes, what are the ranges of the p- and n-doping concentration tuning?

Reply:

We thank the reviewer’s careful examination. **The answer is yes, given that the conductance in p-type and n-type MoS₂ and WSe₂ were switched with large ON/OFF ratio (10^3 - 10^7) by adopting different FE polarization voltage.**

To demonstrate this, we attempted to extract the p/n doping concentration under FE polarization from the measured transfer curves of MoS₂ and WSe₂ using back Si gate after each FE polarization, with n , or $p = \sigma / \mu e$ calculated from the extracted conductance (σ) at $V_{bg} = 0$ and the estimated carrier mobility from $\mu = (L/W) V_{ds}^{-1} C_{gate}^{-1} (dI_{ds}/dV_g)$, where C_{gate} was the gate coupling capacitance with 2D channel. Correct mobility evaluation is then essential for the estimation of carrier concentration.

It should be mentioned that C_{gate} was usually approximated using the oxide capacitance (in our case that for 300 nm SiO₂), this was however based on the assumption of highly conductive semiconductor channel, e.g. the degenerately doped one in our manuscript. In the case of depleted channel, the small semiconductor capacitance start to determine the overall gate coupling. As a result, the mobility can be underestimated, which then leads to the overestimation of carrier concentration in depletion.

To avoid this situation, we adopt the simplified treatment in reference and estimate only those free carriers near band edge that contribute most to the measured conductance in MoS₂ and WSe₂, which is rational since they determine the essential application performance in electronic devices. According to Xiao et al. (Phys. Rev. Lett. 2017, 118, 236801), the free carrier mobility near band edge is not influenced by FE polarization. The free carrier concentration can be then approximated using the band edge mobility, written as $n_{free} = \sigma / \mu_{n,0} e$ for electrons (same for holes).

Here, we approached the band edge mobility in MoS₂ and WSe₂ by the extracted maximum carrier mobility from measured transfer curves under back gate modulation at each polarization. To be specific, for n-doped (p-doped) samples, $\mu_{n,0}$ ($\mu_{p,0}$) was approximated by the maximum mobility extracted at positive (negative) gate bias ($V_{bg} = +30$ or -30 V) that raises the Fermi level close to the conduction (valance) band. The estimated free carrier concentrations in p and n doped MoS₂ and WSe₂ at different FE polarization voltages were now supplied in supplementary material Figure 3. A summary of the tuned carrier concentration range was also given in Table S1.

Supplementary Figure 4. Extracted FE polarization induced free carrier concentration in (a) MoS₂ and (b) WSe₂ after polarization.

Carrier Concentration (cm ⁻³)				
	MoS ₂ (e)	MoS ₂ (h)	WSe ₂ (e)	WSe ₂ (h)
Maximum	5x10 ¹²	10 ¹²	2x10 ¹⁰	3x10 ¹¹
Minimum	10 ⁹	10 ⁹	10 ⁷	10 ⁷

Supplementary Table S1. A summary of the extracted maximum and minimum free carrier concentration in FE-doped MoS₂ and WSe₂.

Given the widely tuned carrier concentration in MoS₂ and WSe₂ by at least 3-4 orders and the large ON/OFF switching in electrical conductance, the method is believed to offer promising doping engineering to various 2D materials, compared to the strategies we have recently summarized (Nanoscale Horiz. 2019, 4, 26): substitutional doping (10¹⁰-10¹² cm⁻²), charge transfer doping (10¹¹-10¹³ cm⁻²). The large remnant ferroelectric polarization is apparent advantageous to conventional oxide dielectrics for the wide electrostatic doping range and reversible pn transition. Compared to the substitutional and charge transfer doping, an essential merit also exists since the present FE doping can be feasibly tuned via the polarization voltage, thus can be feasibly applied for reprogrammable functions.

In the revised manuscript, the discussion on doping range is supplied in Page 7 line 8-14, as following:

“By using the carrier mobility extracted from transfer curves at each FE polarization state, the free carrier concentration tuned by FE polarization was estimated ~10⁹-10¹² cm⁻² in MoS₂ for both electrons and holes, and in WSe₂ ~10⁷-10¹¹ cm⁻² (Supplementary material section II). The reversibly and significantly tuned p/n doping and large ON/OFF switch ratio covering metallic, semiconductor and insulate behaviors will promote their potential applications in various optoelectronic devices with reprogrammable functions.”

3. The following question: The P-V hysteresis is very symmetrical (Supplementary Fig. 1). Can the authors estimate the transition region of doping from P_{up} to P_{down}?

Reply:

We thank the reviewer’s suggestion to estimate the transition region of doping by P_↑ and P_↓. This shall be determined by the transition region of polarization strength for P_↑ and P_↓ by AFM polling. In experiments, because of the electric field distribution near AFM tip, the tip induced

polarization switch decays with the distance from tip position. Also, the previously polarized region may be partially switched by adjacent line scan with reversed bias if in its influence region. Thus, the transition region of doping shall be related to the polarization switch area induced by AFM tip polling, which shall be influenced by the spatial resolution of tip scan or the domain size in FE thin film.

In our experiments, to define FE polarization pattern in device, we adopted scan resolutions of 256x256 or 512x512 for 10x10 μm^2 to 20x20 μm^2 device area, meaning a fine spatial resolution of ~40 nm between adjacent line scans.

On the other hand, to examine the domain size in P(VDF-TrFE) thin film, we performed direct PFM imaging of the domain size formed by applying local polarization using an AFM tip set at 10 V. The results were now supplied in supplementary materials as Figure 4. It was found that local polling at a point lead to a domain size of ~180 nm, while in the case of line polarization by scanning the AFM tip horizontally, the width of the domain size is 100-180 nm. The domain size was thus larger than the spatial resolution adopted in making FE pattern, therefore limited the switching region in forming $P\uparrow$ and $P\downarrow$ patterns.

It was believed that the obtained domain size was not only related to the tip diameter of AFM tip (20-50 nm) applying polarization bias, but also grows with the polarization time (*Sci. Rep. 2013, 4, 4772*) and the thickness of P(VDF-TrFE) polymer (~200 nm) (*J. Phys. Condens. Matter. 2009, 21, 485902*). Thinner FE polymer would enable fine patterning of $P\uparrow$ and $P\downarrow$ at

higher resolution (25-50 nm in 1.78 nm ultrathin films, *Appl. Phys. Lett. 2007, 90, 122904*).

Supplementary Figure 2. PFM image of P(VDF-TrFE) polarized by 10 V AFM tip using (a) line and (b) point scan, from what the FE domain size was estimated ~100-180 nm.

To clarify the resolution by tip bias induced FE polarization, we have included the following sentence in Page 7 line 19-23:

“Note that the expected transition region of remnant polarization ($P\uparrow$ to $P\downarrow$) by switching the

polarity of bias voltage was estimated to be ~100-180 nm (Supplementary material section I), limited by the electric field distribution near AFM tip and the FE domain size in P(VDF-TrFE) at the thickness of 200 nm.”

4. The authors claimed that “FE coupling to 2D semiconductors is favored to realize reconfigurable optoelectronics based on rewritable pn doping in the 2D components, which however hasn’t been demonstrated yet.” (line 57-59 on page3) However, in this manuscript, all MoS₂ devices are not made by the same sample, this means that the authors do not demonstrate the rewritable capability in their configuration either. Can the authors comment on this?

Reply:

We appreciate the reviewer’s careful examination and pointing out this. As the reviewer noticed, we previous adopted two devices throughout the manuscript, one for p, n and pn diode (Figure 1, 2, 3 and supplementary Figure 3), and the other for npn photo-transistor (Figure 4, 5 and supplementary Figure 4). This was indeed because of the long time-span to evaluate each kind of device (n channel, p channel, pn diodes in different pattern designs, and npn transistors).

In response to the reviewer’s comments, in this revision, we have attempted to define pn diode and npn transistor in the same device based on FE polarization. Major modifications to Figure 3, 4, 5 have been made by replacing them with the data obtained on a new device. The updated device displayed a large open circuit voltage ~650 mV and an overall photo-to-electric conversion efficiency ~0.61% as pn diode. When configured into npn photo-transistor, the device manifested responsivity >10 A/W and an ultrafast response speed ~20 μs. **The reconfigured pn diode and npn transistor on the same device thus illustrated the application of rewritable p/n doping in defining devices.**

Since the discussion in Figure 1, 2 focused on the discussion of p/n doping in MoS₂ enabled by FE polarization switching, and Figure 6 intended to validate the high gain observed in npn bipolar transistors, they were kept unchanged in this revision. To avoid confusion, we have added a Table 1 listing all the devices presented in the manuscript by including MoS₂ thickness, width and length in channel and their defined functions.

Table 1. Parameters of MoS₂ devices and their defined functions.

Table 1 Parameters of MoS ₂ devices and their defined functions				
Device	Length (μm)	Width (μm)	Thickness (nm)	Defined Functions
Dev. 1	8.0	7.8	2.4	p, n , pn diode
Dev. 2	7.6	3.2	3.8	pn diode, npn transistor
Dev. 3	6.9	4.7	3.4	pn diode, npn transistor
Dev. 4	7.5	3.0	4.4	nnp transistor

Device parameters of MoS₂ devices studied in this work and their functions demonstrated using FE polarization.

Fig. 3, 4, 5 in manuscript were changed as following, the corresponding discussion were also updated in the main manuscript.

In Figure 3, b-e were replaced with data collected from a new Dev. 2, and the photocurrent map from Dev. 3 in the revised manuscript.

In Figure 4b, PFM image for the device configured into npn phototransistor was replaced with that for Dev. 2 in revised manuscript.

In Figure 5. a-e were all replaced with data collected from new Dev. 2 and the photocurrent map from Dev. 3 in the revised manuscript.

The related discussion of the above figures were all updated in the revised manuscript.

5. About the rectification shown in Fig. 3b, asymmetric I-V characteristics are frequently seen in 2D materials such as WSe₂, MoS₂, etc, due to the Schottky barriers at the contacts. Could the authors comment on how the effects of the contacts are accounted for?

Reply:

We appreciate the reviewer's comment by mentioning the possibility of asymmetric contact in forming the rectification.

To clarify this issue, we performed photocurrent mapping on a pn diode made by FE polarization in the self-driven mode without applying external bias. Since the photocurrent generation relies on successful separation of photogenerated electron-hole pairs in MoS₂, the spatial distribution of photocurrent intensity reflects the local electric field. As indicated in Figure 3f, the photocurrent generation was clearly confined near the junction area defined at the middle of MoS₂ channel (note that the upper region has higher photocurrent response because of thickness change of MoS₂ thin flake outside of the channel). The resulted photocurrent map was clearly different from that Schottky contacted MoS₂ or WSe₂ samples (*Nanoscale*, 2015, 7, 15711, with the photocurrent located near the contact electrode and the resulted photocurrent polarity changes because of the opposite charge separation direction by the Schottky barrier at source and drain terminals), but was similar to MoS₂/WSe₂ PN junction (*Nature Nanotechnol.* 2014, 9, 676). Thus, the mapped photocurrent under zero bias (short circuit condition) validated the rectification from FE polarization defined pn diode rather than the Schottky contacts near electrodes.

To address the same concern from readers, we have provided the photocurrent map in Figure 3f.

Figure 3f A map of self-driven photocurrent in Dev. 3 with pn junction defined in the middle of

channel, compared to its optical microscope image.

The following discussion was included in the revised manuscript at Page 9 line 21-Page 10 line 7.

“To elucidate the self-powered photocurrent generation, a spatial resolved photocurrent distribution has been characterized for a short circuited pn diode defined in Dev. 3. This was achieved by locally illuminating the device using a fine laser spot ($\lambda=532$ nm) in a confocal microscope. The pn junction was defined in the middle of MoS₂ channel. Figure 3f displays the optical microscopy image of the device and the associated photocurrent map. It was clear that most of the photocurrent was generated near the defined junction. We note that self-driven photocurrent may also appear in Schottky contacted devices, but usually with reversed polarity near the source and drain electrodes due to opposite charge separation.⁵⁵⁻⁵⁷ However, in all the devices we studied, photocurrent barely appeared near the contact electrodes and there was no change on the photocurrent polarity across the device area. These results thus validated the role of FE polarization defined pn diode in bringing the high rectification in IV characteristics and self-driven photocurrent.”

6. I am also missing a justification for the current-voltage characteristics found in Supplementary Fig. 2. If that was for gate modulated current in pn diodes at the dark and illuminated condition for different configurations to get qualitative comparisons, it should be mentioned in the paper.

Reply:

We are sorry for the confusion to Supplementary Fig. 2. They were not gate modulated current in pn diode, but corresponded to the current in MoS₂ device under varied FE polarization conditions, with V_p from -20V to 25V.

The intention of the figure was to reflect the contact characteristics in differently doped MoS₂, including the initial n-type, and FE polarization induced heavily n-doped, depleted and reversely p-doped states. In the figure, logarithm scale was adopted in I_{ds} or both I_{ds} and V_{ds} axis to clearly distinguish the different current range at each state, and in Supplementary Fig. 2b to reveal the space charge limited current (SCLC, with $I_{ds} \sim V_{ds}^2$) in p-doped MoS₂ at large bias.

The SCLC behavior was usually observed in insulators or semiconductors with rich trap defects. In present case, both SCLC behavior (supplementary Fig. 3) and Ohmic contact (Figure 3b) were found in p-doped MoS₂ by FE polarization, because of the different hole doping state. In lightly p-doped device I ($G=0.1$ nS at $V_{ds}=0.1V$), SCLC appeared because of the presence of

hole trapping centers within the bandgap, which induced space charge by capturing holes injected from electrodes under large bias. In comparison, for heavily p-doped MoS₂ (device II), the contact was nearly Ohmic because of the above hole trap states were prone to be occupied already given that E_F is closer to valance band.

To avoid this confusion, the discussion on supplementary Fig. 2 was now clearly presented in Page 7 line 25 - Page 8 line 8 together with the discussion of I-V characteristics for p, n doped MoS₂ in revised manuscript.

“For both n and p-doped device, the electrical contact displayed Ohmic behavior due to the degenerate doping in MoS₂. However, we note that for moderately p-doped MoS₂, nonlinear I-V characteristics may appear with ohmic contact at low bias conditions but space charge limited current (SCLC) at large biases, as discussed in Supplementary material section III. Such behavior was however attributed to the charge trapping within the energy gap, which induced non-neutralized space charge in channel upon intense hole injection or under large external bias.⁴⁷ Nevertheless, SCLC behavior was alleviated in heavily p-doped devices by the filling of trap centers when E_F was close to the valance band maximum (VBM).”

Supplementary Figure 5. (a) Current-voltage characteristic of MoS₂ channel after different polling operations with $V_p = 0$ V, -20 V, 25 V, 20 V. (b) displays the plot in log scale, revealing the space charge limited current in p-doped MoS₂ channel under large bias, which indicated the presence of trap filling induced space charges.

Reviewer #2 (Remarks to the Author):

The authors present lateral p-n and n-p-n junctions based on a two-dimensional semiconductor (MoS₂), enabled by local patterning of the ferroelectric polarization in P(VD-TrFE). Using this technique, they realize reconfigurable devices, such as diodes and bipolar transistors, and demonstrate optoelectronic applications such as photovoltaic energy conversion and photodetection.

Although the idea of using ferroelectrics for such purpose is not entirely new, this work constitutes an important advance in this emerging field as the device performance appears to be dominated by the properties of the junctions, rather than the contacts. I thus would like to recommend publication of the manuscript in Nature Communications, provided that the authors can satisfactorily address the following comments/issues:

We sincerely thank the reviewer's comments. All the concerns were addressed as following:

1. In Fig. 3c the authors present photovoltaic properties of their p-n junction. (It is not clear how the data were taken; I assume the whole device was illuminated?) The authors should present the same measurements for n-n and/or p-p configuration to exclude the possibility that the photoresponse stems from the (possibly asymmetric) metal/MoS₂ Schottky junctions. Alternatively, the authors may choose to locally illuminate the p-n junction.

Reply:

We thank the reviewer's suggesting in improving the discussion. Figure 3c was indeed measured by illuminating the whole device. As suggested by the reviewer, we have now added the comparison with the same measurement by doping the whole device area into n or p. The data were presented in supplementary Fig.5. No self-driven photocurrent was observed in the control devices, thus excluding the possible factor of Schottky junctions.

Supplementary Figure 6. Transfer curves of Dev. 2 polarized into (a) n- and (d) p-doped states, its corresponding IV characteristic (b, e) in dark and light illumination conditions, and transient photoresponse to switched light sources, from what it was confirmed that the device displayed no self-driven photoresponse when the whole MoS₂ channel was polarized into n- or p-doped states.

Further, we also attempted to locally illuminate the p-n junction and collected a photocurrent map in short circuit mode, which is now included in the main manuscript as Figure 3f. The results clearly demonstrate that the self-driven photocurrent stemmed from the junction area defined in the middle of MoS₂ channel. Since there were barely photocurrent observed near the contacts, the potential Schottky contact effects can be explicitly excluded.

Figure 3f A map of self-driven photocurrent in Dev. 3 with pn junction defined in the middle of channel, compared to its optical microscope image.

In this revision, we have improved the discussion as following, to account for the potential Schottky contact effect in the observed rectification and self-powered photodetection:

At Page 8 line 24-Page 9 line 2

“In comparison, devices with complete n or p doping across the channel did not display such self-driven photocurrent under illumination (Supplementary material section III).”

At Page 9 line 21-Page 10 line 7

“To elucidate the self-powered photocurrent generation, a spatial resolved photocurrent distribution has been characterized for a short circuited pn diode defined in Dev. 3. This was achieved by locally illuminating the device using a fine laser spot ($\lambda=532$ nm) in a confocal microscope. The pn junction was defined in the middle of MoS₂ channel. Figure 3f displays the optical microscopy image of the device and the associated photocurrent map. It was clear that most of the photocurrent was generated near the defined junction. We note that self-driven photocurrent may also appear in Schottky contacted devices, but usually with reversed polarity near the source and drain electrodes due to opposite charge separation.⁵⁵⁻⁵⁷ However, in all the devices we studied, photocurrent barely appeared near the contact electrodes and there was no change on the photocurrent polarity across the device area. These results thus validated the role of FE polarization defined pn diode in bringing the high rectification in IV characteristics and self-driven photocurrent.”

2. Device operation as bipolar transistor (Fig. 6) is convincingly demonstrated. When operated as phototransistor, though, the device shows some behaviors that resemble those commonly seen in pristine MoS₂ devices: namely, the drop of the photoresponsivity with illumination intensity and the rather slow response time (4 ms). The authors argue by comparison with a pristine MoS₂ device (Fig. 3d) that the response times observed in their junctions are shorter, but 4 ms still seems very long to me. Can the authors provide further evidence that the gain stems from the transistor operation, rather than charge trapping in (short-lived) defects?

Reply:

We thank the reviewer’s careful examination and comments. The bipolar transistor is known as nonlinear photodetector with its gain depends on light intensity. In literatures, one can found the light intensity dependent responsivity in both bipolar transistors based on lateral pnp structured Si nanowire (*Appl. Phys. Lett.* 2016, 109, 033505) and npn structured GaAs/Al_{0.4}Ga_{0.6}As quantum well (*Appl. Phys. Lett.* 1995, 66, 751). The origin of such nonlinearity however was different from the photogate effect usually observed in

photoconductors with trap states (Nano Lett. 2014, 14, 6165; Nano Lett. 2015, 15, 7307) or phototransistors with out-of-plan charge separation (Nature Commun. 2017, 572; ACS Photonics 2016, 3, 2197), instead related to the decrease of injection efficiency at E-B junction (Neamen, D. A., Semiconductor Physics and Devices: Basic principles, 4th edition).

To be clear, the gain in a bipolar phototransistor is due to the amplified electron injection flux from emitter to base compared to the reverse hole flux, $\eta = J_{n,EB}/J_{p,BE}$. In low injection conditions, η is governed by the doping concentration at E and B: $\eta = N_{D,E}/N_{A,B}$ (where, $N_{D,E}$ is the donor concentration in emitter and $N_{A,B}$ is the acceptor concentration in base), which could be made as high as 100-1000 by heavily doped emitter but lightly doped base. However, in the case of large injection conditions, with either a large V_{BE} or I_{BE} , or the high illumination intensity in phototransistor, the injection efficiency decreases due to the increasingly accumulated holes in base (from the separated e-h pairs at reversely biased C-B junction) compared to its inherent doping concentration, with $\eta = N_{D,E}/(N_{A,B} + \Delta p)$. Alternatively, one can consider the increase of recombination loss of injected electron flux from emitter when diffuse across the base region to collector due to the significant accumulation of holes in base.

In usual MoS₂ detectors, the high gain may be contributed by both photoconductive and photovoltaic components (Nano Lett. 2014, 14, 6165). The former originates from the minority carrier trapping into the intrinsic defects of MoS₂ or interface states in device (*Nature Nanotechnol.* 2013, 8, 497), while the latter is related to the out-of-plane charge separation by quantum dots or 2D heterostructures (*ACS Photonics*, 2016, 3, 2197). The gain in both cases decreases with light illumination intensity because of the saturation of charge trapping or photovoltaic separation. **In the following, we clarify that the observed photoresponse gain in device did not origin from such effects, based on the added evidence on fast response ~20 μ s and a photocurrent map of npn phototransistor.**

The photoconductive gain stem from the elongation of recombination lifetime of photogenerated electrons and holes by minority carrier trapping in defects. The resulted $\text{gain} \propto T_{\text{lifetime}}/T_{\text{tr}}$ can be estimated from the extracted photoresponse time and transit time $T_{\text{tr}} = L^2/\mu V$ derived from carrier mobility μ and channel length. In previous, the photoresponse speed (~4 ms) we measured was limited by the speed limitation of light source switching and the speed of measurement unit. Here, by adopting a fast switching LED (365 nm) and a fast measurement unit (B1530, Agilent), we found the response time of npn phototransistor can be as fast as 20 μ s. The photoconductive gain in detector was then estimated to be ~50 considering an apparent mobility of 0.2 cm²/Vs in channel (derived from the measured transfer curve in npn transistor under 0.2 mW/cm²) and the transit time in device ~0.4 μ s, which could not explain the observed gain >1000 in present npn transistor. Moreover, it was found that the

apparent electron mobility increased with illumination intensity (from 0.01 cm²/Vs at 0.02 mW/cm² to 5 cm²/Vs at 39 mW/cm²), thus one expected shorter transit time in device under high illumination intensity. This however contradicted with the reduced photoresponse gain in device under increased light illumination. **Therefore, we could exclude the role of photoconductive gain in dominating the device response.**

On the other hand, the photovoltaic component based on the out-of-plane separation of photogenerated electron-hole pairs (leading to the photodoping of majority carriers in channel) can be fast while offering high gain. To exclude the photovoltaic effect, a photocurrent map of the npn transistor was collected under the bias of 0 and 1V, as shown in Figure 5d in the revised manuscript. It was observed that at $V_{CE}=0V$, photocurrent of reversed polarity appeared near the C and E terminal, which validated the reversed charge separation direction at E-B and B-C junction. At $V_{CE}=1V$, the photocurrent was seen mostly generated near the reversely biased B-C junction, given the same direction of applied electric field and built-in electric field in the space charge region. This is interpreted that the photocurrent contribution in device was determined by the local charge separation efficiency of photogenerated electron-hole pairs. **The photocurrent intensity thus reflected the strength of electric field in device, which matched well with the energy band diagram illustrated in Figure 4c.**

Figure 5d Photocurrent map of the npn bipolar phototransistor under $V_{CE}=0$ and 1V. In right, the lateral distribution of photocurrent was compared to the PFM pattern defining the npn junction.

The obtained photocurrent map also differed clearly from those devices with photogate structure, which usually displayed uniform photocurrent contribution in photogate areas with charge separation or trapping mechanisms. Such difference was indeed related to the gain generation by lateral charge injection or vertical charge separation in devices, since the latter does not lead to apparent spatial photocurrent distribution. Several typical examples we found in literature were listed in the following.

Figure for review only. Photocurrent mapping in various phototransistors with photogate effect dominated gain.

Reference	Materials in Device	Photogate mechanism	Figure
Nano Energy, 2017, 37, 53	BP/WSe ₂	vertical charge separation	a
ACS Nano 2014, 8, 10270.	Graphene/Si	vertical charge separation	b
ACS Appl. Mater. Interfaces, 2018, 10, 36512	ReS ₂	trap in defect states	c

Based on the above evidence, we could exclude the possible photogate effect in the observed photoresponse in npn transistor. The measured photocurrent map also well matched the one expected in lateral npn transistor considering the separation efficiency of photogenerated electron-hole pairs.

In response to the reviewer's comments, we have included the following discussion in the revised manuscript.

At Page 11 line 21-24

“By using a fast switching 365 nm LED source, the photoresponse speed of the bipolar transistor was estimated ~20 μs (inset of Fig. 5a), making it one of the fastest MoS₂ photodetectors but with high gain characteristics.”

At Page 12 line 9-23

“It was noticed R slightly decreases with the negative shift of V_{th} at higher light intensity, which was also usually found in other type phototransistors with photogate effect. However, we

emphasize that the origin of such dependence was different from the usual saturated charge trapping or separation in phototransistor, but due to the increasing recombination losses at the forward biased E-B junction under large injection.²⁸ In Figure 5d, we present the photocurrent map for the present npn bipolar phototransistor under $V_{CE}=0$ and 1V to validate its operation principle. It was seen that photocurrent of reversed polarity appeared near C and E terminal at $V_{CE}=0V$, while at $V_{CE}=1V$ the photocurrent was more efficiently generated near the reversely biased B-C junction. This was consistent with the expected electric field strength in device that eventually separate the photogenerated electron-hole pairs. The present photocurrent map also differed from other type phototransistors that usually displayed uniform photocurrent distribution within the photogate area.⁶⁰⁻⁶² Such difference in the two kinds of phototransistors was indeed ascribed to their different gain generation mechanism in device, *i.e.* via the lateral in-plane charge injection and the out-of-plane photovoltaic effects, respectively.”

- It would also be helpful if the authors could comment on the stability of the junctions (versus time, bias voltage, etc.)

Reply:

We thank the reviewer’s comments. Though FE coupled devices can in principle exhibit long retention characteristics by the bistable polarization states of FE materials, the practical device experience instability issues due to the incomplete compensation of the depolarization field in FE component (Phys. Rev. Lett. 1973, 30, 1218; Mater. Today 2011, 14, 592). In present work, instability of FE polarization occurs after the polling by AFM tip, since the depolarization field could not be screened without metal contacts at the top surfaces of FE component. However, given the freedom in defining local polarization pattern, the FE polling using AFM tip is suitable for investigating devices of various configurations (Phys. Rev. Lett. 2017, 118, 236801) without concerning the material uncertainties in differently configured devices, as the case we have presented in the manuscript.

In experiments, we observe that the device after AFM polling tended to degrade in 1-2 hours, without apparent dependence on the applied bias but can be accelerated under thermal effect. In the following figure, MoS₂ conductance after P/N doping by FE polarization was seen to degrade, but after 1h still maintains the defined P/N doping type. The study reported here were generally conducted within a short period before significant degradation occurred. An example of the degradation for p and n-doped MoS₂ was shown in following.

Figure for review only. a) The degradation of MoS₂ conductance after p- and n-type doping by FE polarization, and b, c) the transfer curves of p- and n-doped MoS₂ right after FE polarization and 4000s later.

We note that such instability could be feasibly avoided by adopting the top-gate structure, which had been widely demonstrated in the past as either memories (Nano Lett. 2016, 16, 334; Adv. Mater. 2012, 24, 3020;) or photodetectors (Adv. Mater. 2015, 27, 6575). Predefined multiple top-gate terminal shall be easily integrated to enable a reconfiguration of device function after manufacturing.

To be clear to the readers, we have in this revision included the following comments on the stability of device and the efforts necessary in the future.

At Page 5 line 4-7

“Despite the switched FE polarization tended to relax due to the incomplete compensation to depolarization field in P(VDF-TrFE), it enables rewritable polarization pattern on the same device, thereby allowing the direct study of the influence of device configurations without worrying material differences.”

At Page 14 line 22-Page 15 line 2

“Further maturation of such strategy toward array-structured functional optoelectronic devices with high stability shall be viable based on predefined top-gate patterns or electrical imprint methods.”

At Page 15 line 22-Page 16 line 2

“After AFM polling, the FE polarization in device tended to degrade within 1-2 hours because of the lack of screening to depolarization field without top metal contacts. The device performance was therefore studied within this period, out of the period the device was repolarized at the same conditions for further measurements.”

Reviewer #3 (Remarks to the Author):

In this work, the authors report some interesting results on the integration of MoS₂ with ferroelectric materials. The authors demonstrated reconfigurable photodetectors with high response. Overall the results are interesting but I am not convinced completely that this work should be published in Nature Communications.

1. It is not clear why such reconfigurability is needed. We can easily apply an electric bias to tune the device operational condition. Moreover, there are many different approaches to make photodetectors and the introduction of ferroelectric materials into optoelectronic devices do not seem to be well-justified, at least in this case. You can make a pn junction or directly make a phototransistor. You can also make an APD. Sometimes you do need reconfigurable photonic devices (e.g. in optical networks) but I do not think here making a reconfigurable photodetector has intrinsic advantages.

Reply:

We thank the reviewer's critical comments, which helps us to improve the work. It is undoubted that the reconfigurability renders much freedom in defining the device function or performance after manufacturing. For photodetectors, the essential requirements include fast response, high gain, and low energy consumption in arrayed image sensors. This was usually fulfilled in market by different kinds of photodetectors, e.g. diodes, transistors, or APD, depending on the target application in sensing, imaging or optical communication. It was however difficult to balance the response speed, gain and energy consumption in complicated scenes, e.g. the one perceived by human vision system with dramatically varied light conditions.

The photodiode is the most energy efficient type of detector because of its self-driven operation, whereas its low gain <1 make it more suitable for high light levels. It had been earlier proposed that the photodiode in image sensor can be used for energy harvesting in daytime with high light illumination conditions (Khondker Ahmed et al., *Reconfigurable 96x128 Active Pixel Sensor with $2.1\mu\text{W}/\text{mm}^2$ Power Generation and Regulated Multi-Domain Power Delivery for Self-Powered Imaging*, ESSCIRC Conference 2016, DOI:10.1109/ESSCIRC.2016.7598352; Chao Shi, et al. *A CMOS Image Sensor with Reconfigurable Resolution for Energy Harvesting Applications*, IEEE Sensors Conference, 2009). On the other hand, bipolar transistors could offer high gain for the detection at low light levels but required more energy due to dark current issues. Reconfigurable switching between pn diode and npn bipolar transistor in an image array would however balance their performance for varied light level conditions in environment. For example, in an image sensor, pn diode can be configured in day time for imaging and

energy harvesting, while at night time, the npn transistor could be made for imaging at low light levels. The direct reconfiguration of detectors themselves allows switching of imaging capability without changing the whole image sensor array or optical paths. Considering that human visual system also consists of different neurons that can adapt to different light levels, the reconfigurability in FE polarized photodetector may be potentially used for developing smart image sensors.

In the revised manuscript, the following sentences have been included to elucidate the potential merits of having reconfigurability in optoelectronic devices.

Page 2 line 10-11

“An ultimate pursuit to this end would be however a reconfigurable function device that can be customized on demand, so that a universal device architecture can be deployed in various application scenes.”

Page 3 line 22-24

“Such reconfigurable device characteristics may promote the evolvement of smart image sensors that reflect to external light environments for the balanced photoresponse gain and energy efficiency.”

2. The performance of the photodetector is not very impressive. Indeed the responsivity is high due to the gain. However the response time is long (4 mS) and as a result, *the speed is very low (below kHz)*. It is very easy to achieve high responsivity if you do not care about the speed. The difficult part is high responsivity, high speed and low noise simultaneously. In fact, such a high gain can be easily achieved in a simple silicon photoconductor if high speed is not needed.

Reply:

We thank the reviewer’s critical comments. The npn transistor could be potentially operated at a fast response speed and high gain. In our case, the optical gain >1000 was so far the highest in various kinds of bipolar transistors based on 2D materials, whereas due to the speed limitation of light source and measurement unit, the photoresponse speed in the configured npn transistor was previously underestimated. In this revision, we have performed another measurement of the photoresponse speed by using a fast switching LED (M365FP1, Thorlabs) and a fast measurement unit (B1530, Agilent). We found the npn phototransistor exhibited a fast photoresponse <20 μ s, making it one of the fastest photodetectors based on 2D materials while offering the high gain factor, as indicated in the following Table S2. The results could therefore demonstrate the high performance of the present npn phototransistor, and the

potential of exploiting FE polarization enabled p, n doping in 2D materials and further patterned doping for function device applications.

In this revision, an inset figure was added in Figure 5a showing the fast transient response, and in Figure 5f the responsivity and speed were updated with the new data collected from the npn phototransistor on device II, thereby demonstrating the fast photoresponse capability of npn transistors.

Figure 5. Inset of Figure 5a, transient photoresponse of the npn phototransistor to fast switching 365 nm light illumination, indicating a response time constant $\sim 20 \mu s$. **e** Comparison of the photodetection performance of npn bipolar phototransistor with other MoS₂ devices defined this work and found in literatures, including photoconductor (square), photodiode (diamond) bipolar phototransistor (circle) and other type phototransistors (triangle).

Supplementary Table 2.

Table S2 performance comparison of MoS ₂ photodetectors in different device configuration					
Configuration	Materials	λ (nm)	Speed (s)	R ($A W^{-1}$)	Ref.
Photoconductor	MoS ₂	561	20	880	3
	MoS ₂	635	0.01	10	18
Photodiode	MoS ₂ /Lateral junction	575	0.75	0.308	63
Phototransistor	MoS ₂ /Top gate	532	7×10^{-5}	0.57	64
	MoS ₂ /Vertical pn junction	635	0.01	7.8	65
Bipolar transistor	MoS ₂ -BP	1550	1.5×10^{-5}	0.153	34
	MoS ₂ npn	532	2×10^{-5}	11.9	This work

References in Table S2.

- Lopez-Sanchez, O., Lembke, D., Kayci, M., Radenovic, A. & Kis, A. Ultrasensitive photodetectors based on monolayer MoS₂. Nat. Nanotechnol. 8, 497-501 (2013).
- Kufer, D. & Konstantatos, G. Highly Sensitive, Encapsulated MoS₂ Photodetector with Gate

Controllable Gain and Speed. Nano Lett. 15, 7307-7313 (2015).

34. Li H, Ye L, Xu J. High-Performance Broadband Floating-Base Bipolar Phototransistor Based on WSe₂/BP/MoS₂ Heterostructure. ACS Photonics 4, 823-829 (2017).

63. Zhang, X. et al. Poly(4-styrenesulfonate)-induced sulfur vacancy self-healing strategy for monolayer MoS₂ homojunction photodiode. Nat. Commun. 8, 15881 (2017).

64. Tsai, D. S. et al. Few-Layer MoS₂ with high broadband Photogain and fast optical switching for use in harsh environments. ACS Nano 7, 3905-3911 (2013).

65. Huo, N. & Konstantatos, G. Ultrasensitive all-2D MoS₂ phototransistors enabled by an out-of-plane MoS₂ PN homojunction. Nat. Commun. 8, 572 (2017).

Modification to the manuscript includes:

At page 11 line 21-24.

“By using a fast switching 365 nm LED source, the photoresponse speed of the bipolar transistor was estimated ~20 μs (inset of Fig. 5a), making it one of the fastest MoS₂ photodetectors but with high gain characteristics.”

At Page 13 line 7-11

“As indicated in Fig. 5e, when compared to other MoS₂ photodetectors in either photoconductors,^{3, 18} photodiode⁶³ or the phototransistor configurations,^{34, 64, 65} the high gain value here still gave rise to competitive photodetection performance by delivering simultaneously the fast speed response speed and a high responsivity.”

At Page 16 line 4-6

“To probe the photoresponse speed, a 365 nm fast switching LED (M365FP1, Thorlabs) and fast measurement unit (B1530, Agilent) was used.”

Finally, we sincerely thank all the reviewers' precious comments again, which helped us to solidify the work from many aspects.

REVIEWERS' COMMENTS:

Reviewer #1 (Remarks to the Author):

The new manuscript has been well revised according to the reviewers' comments. In particular, they have detailed the rewritable capability that was lacking in the original manuscript and have provided experimental evidence by adding different devices. As a result, the paper is in much better shape now. Therefore, I suggest that this revised manuscript is now suitable for publication.

Reviewer #2 (Remarks to the Author):

The authors have fully addressed my remarks and have significantly improved the quality of the manuscript. I recommend publication of the manuscript in Nature Communications.

Reviewer #3 (Remarks to the Author):

The authors addressed some of the concerns but the referee here still feels this work does not represent an important breakthrough in nanophotonics. First, the speed is still very low. 20 μ s response time, in fact, indicates that the device can only operate in tens of kilohertz. Such a speed is not impressive at all even with the gain of 1000. If we use the gain-bandwidth product as the performance metric, an APD can easily operate at a gain of 100 and speed in GHz range (ns response time). The gain-bandwidth product is at least 3 orders of magnitude better for APD (it could even be 4 orders of magnitude better).

Second, modern optoelectronic devices are usually made from semiconductor heterostructures for optimal performance. The referee here does not think such gate-induced, homojunction pn diodes and pnp (or npn) transistors can have potential for practical applications.

Reply to reviewers' comments (NCOMMS-18-32079A)

Reviewer #1 (Remarks to the Author):

The new manuscript has been well revised according to the reviewers' comments. In particular, they have detailed the rewritable capability that was lacking in the original manuscript and have provided experimental evidence by adding different devices. As a result, the paper is in much better shape now. Therefore, I suggest that this revised manuscript is now suitable for publication.

Reply: We sincerely thank all the comments from the reviewer that helped to greatly improve our manuscript.

Reviewer #2 (Remarks to the Author):

The authors have fully addressed my remarks and have significantly improved the quality of the manuscript. I recommend publication of the manuscript in Nature Communications.

Reply: We really appreciate all the reviewer's comments that instructed us to improve the discussion and quality of the manuscript.

Reviewer #3 (Remarks to the Author):

The authors addressed some of the concerns but the referee here still feels this work does not represent an important breakthrough in nanophotonics.

1. First, the speed is still very low. 20 μ s response time, in fact, indicates that the device can only operate in tens of kilohertz. Such a speed is not impressive at all even with the gain of 1000. If we use the gain-bandwidth product as the performance metric, an APD can easily operate at a gain of 100 and speed in GHz range (ns response time). The gain-bandwidth product is at least 3 orders of magnitude better for APD (it could even be 4 orders of magnitude better).

Reply:

We appreciate the reviewer's critical comments to the manuscript, which have pushed us to improve the performance of detector. By improving the sampling rate in measurements, we observe the best photodetection speed for the npn photodetector was \sim 3-5 μ s, which at

present stage shall be the switching limit in our experiments setup (Supplementary Figure 11). However, in principle, the speed of bipolar transistor could be optimized by suppressing extrinsic defects in MoS₂ and by refining the width and length of base and collector region, so to minimizing the effect of charge trapping and parasitic capacitance in response speed. Having higher mobility in MoS₂ shall also contribute to faster response by reducing the carrier transit time in base. [*High-Speed Electronics and Optoelectronics: Devices and Circuits*, Cambridge University Press, 2009]

Supplementary Figure 11. Transient photoresponse of npn diode at $V_{ds}=5$ V to fast switching light illumination at 365 nm. The measurement is conducted using a sampling interval of 1 μs and the response time is estimated ~3-5 μs.

When comparing the performance of photodetectors of different types, one need to admit that there is no one kind of detector that fits for every application, e.g. for image sensors, optical communication, photon counting, etc., due to their different requirements in sensitivity, speed, linearity. Specifically, though APD (avalanche photodiode) exhibit large gain while providing fast response, it requires large operation voltages to trigger impact ionization under intense electric fields ($>3 \times 10^5$ V/cm for Si) in reversely biased photodiode. As the result, Si APDs usually operates at high voltages ~150V for gain >100 , while for single photon detection in Geiger mode, even higher voltage above the breakdown voltage is necessary.[*Hamamatsu opto-semiconductor Handbook*] For MoS₂, the required electric field to trigger impact ionization is also as high as 5×10^5 V/cm.[*ACS Nano*, 2018, 12, 7109-7116] Besides, the impact ionization in APDs will not stop instantly when switching off the light excitation, but will rely on the drop of reverse bias on the active junction below an ionization threshold. Therefore, for practical high speed photodetection, it would require complicated external voltage control circuits, including high voltage supply and feedback circuits.[*Hamamatsu opto-semiconductor Handbook*] Hence, though APDs exhibit excellent performance in high end photodetectors, its high density integration is not as easy as other type photodetectors, including the reported pn

junction and bipolar transistors in present work.

In terms of the response speed and gain, the present bipolar phototransistors yield competitive performances at considerably lower operation voltages, which may benefit its application in imaging and wearable devices.

To clearly explain their difference, the following discussion have been included in the main manuscript:

On Page 10 line 10-12

“Compared to the avalanche photodetector (APDs), the bipolar phototransistor could work at considerably lower operation voltages (~150 V for commercial Si APDs) while yielding the similar photodetection gain.”

On Page 10 line 25- Page 11 line 5

“Faster response within as short as ~3-5 μs is also achieved in experiments (Supplementary Figure 11), which is close to the switching limit of the adopted light source. It is believed that the ultimate device response speed depends on both material characteristics and device geometries. Further improved speed is likely attainable given higher carrier mobility in MoS_2 and improved design on the width of base and collector, as they directly determine the overall carrier transit time in device.

2. Second, modern optoelectronic devices are usually made from semiconductor heterostructures for optimal performance. The referee here does not think such gate-induced, homojunction pn diodes and pnp (or npn) transistors can have potential for practical applications.

Reply:

We appreciate the reviewer’s comments. Heterojunction bipolar transistors (HBT) principally yield higher gains than homojunctions by having a wide band gap semiconductor as the emitter compared to base with narrower bandgap. However, the high performance comes at the price of the efforts in optimizing the junction interface, as good lattice matching and high-quality interface are always necessary to reduce electron-hole recombination loss there. Therefore, the present HBT markets are dominated by III-V semiconductors (InP, GaAs, AlGaAs, GaN) and SiGe heterojunctions. However, it should be mentioned that the homojunction bipolar transistor could offer well balanced sensitive and speed with considerably reduced material and fabrication cost. Again, it shall have position in

photodetector market when cost is concerned.

In addition, the present bipolar transistors based on the ultrathin 2D van der Waals materials may benefit flexible detectors compared to the traditional devices based on vertical epitaxy layers by avoiding strain issues.[*Nature Communications*, 2018, 5266] It is therefore believed that 2D homojunction bipolar transistor can be practical by fulfilling specified several but not all requirements.

As the reviewer has mentioned, hetero-structured bipolar transistors are promising for better performances. This can also be done based on artificially assembled 2D semiconductors, whereas at present the limited control on the exfoliation and stacking of 2D materials, and also doping state in each region (emitter, base and collector) make it challenging to achieve optimal device performances. With the ultrathin thickness (<10 nm) of 2D materials comparable to the Debye length, gate modulations were often employed to enhance the junction characteristics and optimize the performance of various kinds of 2D devices, the same however cannot be applied for conventional 3D semiconductors with large characteristic size. In present work, the adopted ferroelectric gate modulation not only enables widely tuned doping polarity and carrier concentration over conventional gate oxides, the reconfigurable polarization pattern here also greatly facilitates the exploration of high-performance photodetectors by feasibly changing the design, which have not been possible before. The device configuration thus is also not limited to the already demonstrated pn junction and bipolar transistors, but includes APDs mentioned by the reviewer. Further exploration of ferroelectric coupled photodetector in different designs would therefore undoubtedly promote the evolution of high performance 2D photodetection.

Accordingly, we have included the following discussion that outlooks the possible future efforts toward higher performance photodetection.

On page 15 line 1-4

“The gate-free yet reconfigurable methodology introduced the great potential of exploiting locally coupled FE polarization in customizing high performance optoelectronic devices based on the thriving 2D semiconductors and in the future their van der Waals heterojunctions, which in principle could offer even larger speed and gain product than present homojunctions.

Finally, we sincerely thank all the reviewer’s comments that pushed us to improve the discussion and quality of the manuscript.